# Environmental microbiome diversity and stability is a barrier to antimicrobial resistance gene accumulation
Uli Klümper [1,13], Giulia Gionchetta[2,13], Elisa Catão[3,4,13], Xavier Bellanger[3], Irina Dielacher[5], Alan Xavier Elena[1], Peiju Fang[1], Sonia Galazka[6], Agata Goryluk-Salmonowicz [7,8], David Kneis [1], Uchechi Okoroafor[9], Elena Radu[5,10], Mateusz Szadziul [7], Edina Szekeres[11], Adela Teban-Man[11], Cristian Coman[11], Norbert Kreuzinger[5], Magdalena Popowska[7], Julia Vierheilig [5,12], Fiona Walsh[9], Markus Woegerbauer[6], Helmut Bürgmann [2,14], Christophe Merlin[3,14] & Thomas Ulrich Berendonk [1,14] ✉

When antimicrobial resistant bacteria (ARB) and genes (ARGs) reach novel habitats, they can become part of the habitat's microbiome in the long term if they are able to overcome the habitat's biotic resilience towards immigration. This process should become more difficult with increasing biodiversity, as exploitable niches in a given habitat are reduced for immigrants when more diverse competitors are present. Consequently, microbial diversity could provide a natural barrier towards antimicrobial resistance by reducing the persistence time of immigrating ARB and ARG. To test this hypothesis, a pan-European sampling campaign was performed for structured forest soil and dynamic riverbed environments of low anthropogenic impact. In soils, higher diversity, evenness and richness were significantly negatively correlated with relative abundance of >85% of ARGs. Furthermore, the number of detected ARGs per sample were inversely correlated with diversity. However, no such effects were present in the more dynamic riverbeds. Hence, microbiome diversity can serve as a barrier towards antimicrobial resistance dissemination in stationary, structured environments, where long-term, diversity-based resilience against immigration can evolve.

The spread of antibiotic resistance genes (ARGs) represents one of the biggest challenges to future human and animal health[1,2]. It has become clear that tackling the issues brought on by the expansion of antimicrobial resistance (AMR) requires a global effort that spans the fields of human, veterinary, and environmental health[3,4]. This perspective, also known as the "One Health" approach, is frequently used as a framework for worldwide efforts to contain the spread of AMR. However, integrating the environmental sphere has, due to its high complexity, proven to be particularly challenging. Understanding the existing ecological dispersal barriers is necessary to restrict the propagation of AMR in the environment[5–7].

ARGs are ancient, naturally occurring in environmental bacteria and evolved over billions of years[8]. However, during the last century, environmental microbial communities have been constantly subjected to invasion events by antibiotic resistant bacteria (ARB), and their associated ARGs, which have been enriched or released through anthropogenic activities. For example, release and reuse of wastewater effluents or the application of manure to soils are known to be primary conduits of AMR spread in the aquatic and terrestrial environments[9–12]. But even habitats with no direct anthropogenic impact are regularly exposed to lower frequencies of such invasion events, for example, through wildlife and aerial depositions. These natural diffuse sources of AMR-pollution promote the widespread dispersal of AMR over time at low abundances[13–17].

In the case when ARG or ARB are enriched and introduced through anthropogenic actions and immigrate into a natural environmental community, the term invasion is more exact, although similar ecological principals as in immigration processes apply. Invasion has been defined, both on the micro- and the macro-biological scale, as a process consisting of a sequence of successive steps, namely, (i) introduction, (ii) establishment, (iii) growth and spread, and (iv) impact[18]. For it to be successful, the invader has to overcome the "biotic resistance" towards invasion of the habitat[19]. In theory, this process becomes more difficult with increasing biodiversity[20]. The capacity to exploit the resources provided by a habitat is enhanced when communities exhibit a greater diversity, which in turn reduces opportunities for invaders, hence lowering their persistence[21]. If an invader is phylogenetically, thus presumably also physiologically, close to established community members occupying its specific niche, the invasion process has been

shown to become merely stochastic[22]. In contrast, even small-scale disturbance events, such as exposure to a novel stressor or changes in temperature, can considerably increase invasion success by affecting the niche occupancy of resident species[23–25]. However, such effects strongly depend on the niche partitioning within the given environment rather than diversity alone[26], and might hence be far more pronounced in structured environments with a long-term established niche occupation.

In the context of AMR dissemination, it is reasonable to assume that ARB reaching a natural community may persist longer when the biodiversity of the autochthonous community is low, as this usually coincides with a lower degree of niche occupation[21]. This could result in a longer-term establishment of the ARB, and thus increase the relative abundance of ARGs. Even when the e process is not successful in the long term, a slightly prolonged residence time of the invader could have a significant impact on the likelihood of ARGs being horizontally transferred to the endemic microbiota[27]. Alternatively, future invasion events could be favored by reducing the community resilience, through, for example, reduction of microbial network connectivity and hence the competitiveness of the indigenous community[28]. Reciprocally, high microbial diversity could impede the spread of ARB and ARGs, thus serving as a potential natural barrier.

Diversity as a limiting factor for AMR invasion was demonstrated in the short-term for laboratory soil microcosms inoculated at different diversities using an artificial dilution-to-extinction approach. Less diverse soil microcosms displayed a far higher likelihood of being invaded by ARBs[29]. Regarding the aquatic dimension, increased invasion success of a resistant *E. coli* strain into river biofilm communities under stress conditions was shown to coincide with a loss in diversity of these communities[24]. Further, diverse microbial communities with a high degree of functional niche coverage, such as activated sludge, have been suggested to provide natural barriers for the proliferation of AMR[30]. However, data suggests that activated sludge communities have also incorporated a particularly high diversity of ARGs encoded on mobile genetic elements (MGEs)[31], as abundant resources and high proximity due to high bacterial densities promote horizontal gene transfer of mobile ARGs[9].

Based on this theoretical and experimental knowledge, we hypothesize that the pervasiveness of AMR into environmental microbiomes is inversely correlated to the diversity of the communities in question. To our knowledge, no prior field-based study has explored whether AMR dissemination is indeed related to microbial diversity in terrestrial or aquatic environmental microbiomes. To this end, low impacted environmental samples with no direct point or non-point anthropogenic depositions through solid or liquid waste discharge were collected. Their ARG diversity and abundances, beyond natural background levels, are hypothesized to result from the accumulation of invasion success of ARB and ARG over time introduced through the previously mentioned dispersion routes. Therefore, ARG diversity and abundance would rise, if after a successful introduction, ARB were able to either establish themselves long-term in the indigenous microbial communities or transferred their mobile ARG load. Consequently, 167 of such low anthropogenic impact environmental samples were collected during fall/winter 2020/21 across seven European countries. Half of these were taken from forest soils, representing a stationary, structured environment, while the other half were obtained from river sediments and biofilms representing a more dynamic environment. For each sample, the bacterial diversity was assessed through 16 S rRNA gene sequencing. The resistome, defined as the collection of all ARGs in a microbiome, was analyzed via abundance of 27 clinically-relevant ARGs determined through high-throughput chip-based qPCR. Simultaneously, the abundance of mobile genetic elements (MGEs) in samples was assessed through 5 marker genes regularly associated with AMR and the anthropogenic fecal pollution indicator crAsssphage[32,33] was quantified. This allowed to ultimately test the previously presented hypothesis that AMR in low-anthropogenic-impacted environmental microbiomes is inversely correlated to the diversity of the

communities in question as microbiome diversity can serve as a barrier against the spread of ARGs.

## Results

### Assessing diversity in the river and soil dataset

Two complementary sets of samples of low anthropogenic impact were obtained from a total of 94 riverbed (61 river epilithic biofilm and 33 river sediment samples) as well as 73 soil samples. When assessing the beta diversity of the bacterial communities, no clear distinction was observed comparing sediments and epilithic biofilms (PERMANOVA, pseudo-F = 2.77, p > 0.05). Consequently, these samples were subsequently grouped to create the combined river dataset. Soil samples differed significantly and with a large effect size from those obtained from river samples (PERMANOVA, pseudo-F = 14.73, p < 0.001) (Fig. 1a).

The aquatic samples contained 13,363 individual, bacterial ASVs (Supplementary Data 1) which accounted for 19 phyla with an average relative abundance above 1% and were, throughout, dominated by bacteria belonging to the phyla *Proteobacteria*, *Bacteroidota* and *Actinobacteriota* (26 ± 9%, 17 ± 13%, 12 ± 9%) (Supplementary Fig. 1). In the soil dataset 13,451 individual bacterial ASVs (Supplementary Data 2) were detected and the phyla *Acidobacteria*, *Actinobacteriota* and *Proteobacteria* (18 ± 4%, 15 ± 2%, 13 ± 3%) dominated (Supplementary Fig. 2). CrAssphage, a common indicator for recent anthropogenic fecal pollution[32,33], was undetected in the entirety of soil samples and in 78% of the river samples. For the latter it remained at low relative abundance below $10^{-5}$ copies per copy of the 16 S rRNA gene, confirming that samples were indeed of low anthropogenic impact origin. For the three main alpha-diversity metrics - Chao1 richness, Shannon diversity and Pielou evenness - high and low biodiversity samples for each of the two datasets, rivers and soils, were obtained (Fig. 1b, Supplementary Data 3 & 4). The main distinction between the two datasets was the significantly higher level of Pielou evenness in the dataset from the structurally stable soil (0.95 ± 0.02) compared to the dynamic river environment (0.89 ± 0.08) (p < 0.0001, f = 48.78, One-Way ANOVA). The differences obtained subsequently allowed these diversity metrics to be used as test variables for correlation with ARG abundance.

### Resistome diversity and abundances

To analyze the resistome of the soil as well as the river samples, we performed high-throughput qPCR of 27 ARGs together with the 16 S rRNA gene to obtain relative abundances of the ARGs for each sample. On average, 18.44 ± 5.61 of the 27 tested ARGs were successfully detected in samples from the river dataset. Slightly, but significantly, less ARGs per sample (15.95 ± 6.05; p = 0.014, f = 6.11, One-Way ANOVA) were successfully detected in the soil dataset. In both resistome datasets, the *aac*(3)-VI gene conferring aminoglycoside-resistance was the most abundant (Fig. 2). In the river dataset, genetic determinants for sulfonamide (*sul*1), vancomycin (*van*A), colistin (*mcr*1) and phenicol (*flo*R) resistance clustered together as dominant, followed by other ARGs that promote resistance to macrolides, lincosamides and streptogramins B - (*mph*A), β-Lactam (*bla*$_{CTX-M2}$, *bla*$_{CTX-M1}$, *bla*$_{CMY2}$), phenicol (*cml*A) and aminoglycoside (*aac*(6')-lb3, a*ph*(3')-Ib) antibiotic classes. ARGs conferring resistance to quinolone, trimethoprim and tetracycline were less abundant, although in certain soils, particularly from Poland and Romania, the corresponding abundance of individual genes (e.g., *tet*(W), *qnr*S, *drf*A) was higher (Fig. 2a). In general, considerable clustering of samples was observed. The Irish samples clustered together with a few Romanian and one German sample separately from the rest, and displayed an overall lower relative ARG abundance with the exception of the *bla*$_{TEM}$ gene, which displayed a particularly high abundance in Ireland (Fig. 2a).

In the soil dataset, the *aac*(3)-VI (aminoglycoside), *dfr*A1 (trimethoprim), *mph*A (MLS$_B$) and *qnr*S (quinolone) genes clustered together as the most abundant determinants in most countries (Fig. 2b). ARGs conferring resistance to colistin (*mcr*-1), phenicol (*cml*A2), β-Lactams (*bla*$_{CMY-2}$, *bla*$_{CTX-M}$) and aminoglycoside (*aph*(3')-Ib,

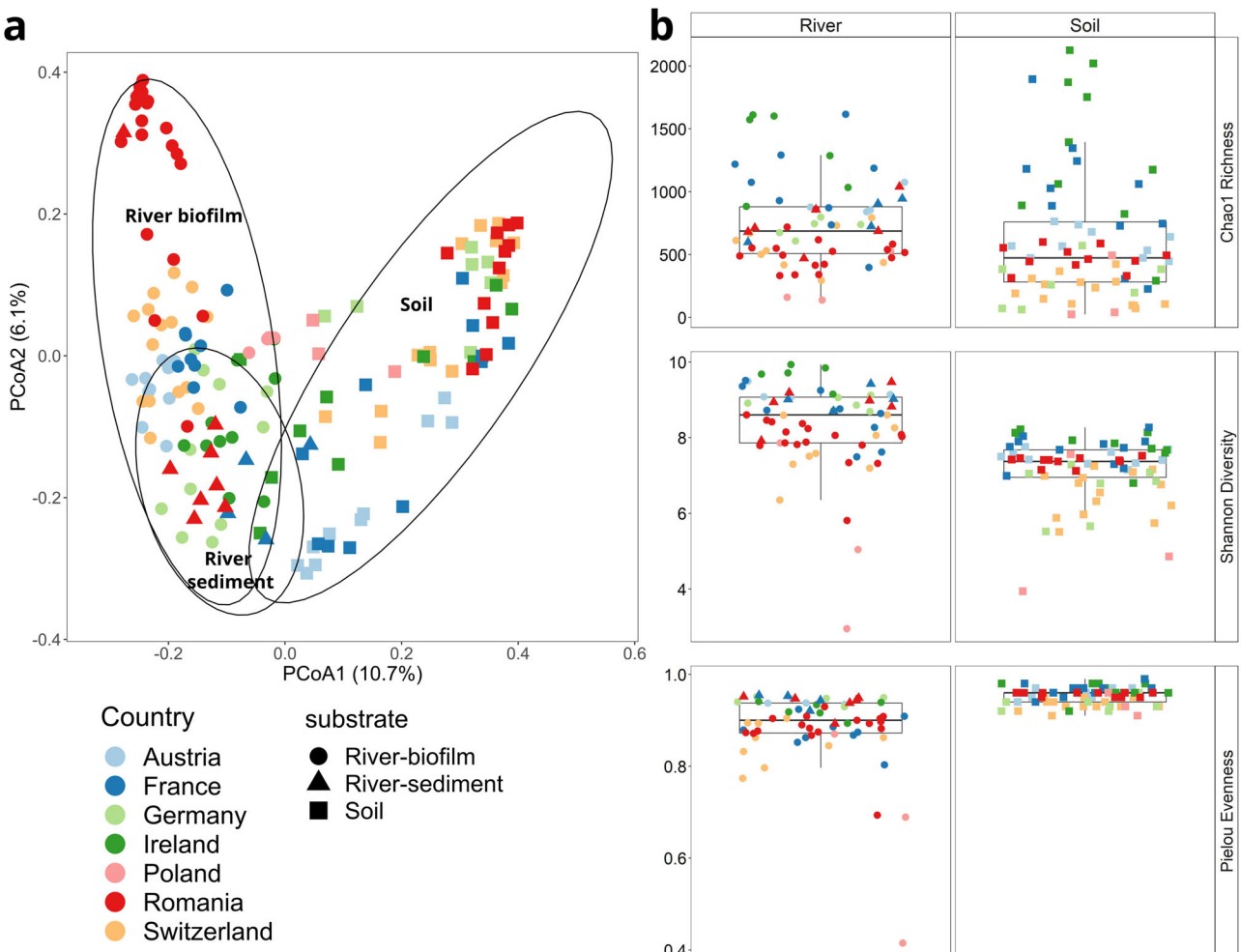

**Fig. 1 | Diversity of the river and soil datasets. (a)** PCoA of the beta diversity based on Bray Curtis distance of ASV relative abundance data from riverbed materials (sediments: triangles & biofilms: circles) and soil (squares). Colors code for the country of origin. Ellipses were drawn based on a 95% confidence interval to represent samples from each of the substrates. **(b)** Alpha-diversity indices (Chao1 richness, Shannon diversity and Pielou evenness) from riverbed materials and soil collected from the seven countries.

$aac(6')$-lb3) were detected in most countries at intermediate abundances, but with highest values in Switzerland and France. Accordingly, the Swiss and French soil resistomes clustered together as most ARGs were found at higher and similar abundances compared to the other countries (Fig. 2b). Irish soils displayed again the lowest number of ARGs detected and similar to the river dataset the abundance of $bla_{TEM}$ was significantly elevated in Ireland compared to other countries (Fig. 2b). The colistin ARG $mcr-3$ was the sole ARG below the limit of detection for all samples, and therefore not used for further analysis.

**Identification of potential ARG hosts**

As a first step of connecting diversity within the samples with the ARG content we performed correlation analysis between the obtained ASV and ARG abundances to identify if, within each dataset, certain ARGs can clearly be attributed to single or multiple bacterial hosts and would hence potentially be independent of overall community diversity. However, in our datasets spanning geographical distances, no clear host identification was possible based on correlation analysis: In the soil dataset, each ARGs abundance significantly positively correlated on average with 272 ± 260 ASVs, based on Pearson correlation of abundances with Bonferroni correction for multiple testing (Supplementary Data 2). For 21 of the 25 detected ARGs the number of positively correlated ASVs exceeded 20 and reached up to 984 correlated ASVs for $dfrA$, while for two of the ARGs ($aac(3)$-VI, $aph(3')$-Ib) not a single correlated ASV could be identified. Only

for the ARGs $bla_{CTX-M2}$ a single ASV (classified as *Acidobacteria* subgroup 2) and for $bla_{OXA48}$ two individual ASVs (classified as *Acidothermus* & *Xanthobacteraceae*) were correlated positively as potential main hosts (Supplementary Data 2). Similarly for the river dataset (Supplementary Data 1), each ARGs abundance significantly positively correlated with 101 ± 88 individual ASVs and exclusively for $vanA$ only two potential host ASVs (classified as *Saprospiraceae* and *Sphingobacteriales* AKYH767) were identified through correlation analysis (Supplementary Data 1). However, neither of the potential hosts identified for $bla_{CTX-M2}$, $bla_{OXA48}$ in soil and $vanA$ in rivers have previously been reported in the literature as hosts of these ARGs, meaning that in this case correlation is likely not associated with causation. Metagenomic analysis with contig assembly and network analysis, as regularly carried out for anthropogenically impacted environments with high ARG abundance and lower bacterial diversity[31,34], might be able to provide a higher resolution of exact hosts. However, taking into account the low detection limit of metagenomics, the high diversity of potential host bacteria and generally low relative abundance of individual ARGs in these low impacted environmental samples, the necessary sequencing depth and coverage to conclusively capture the potential ARG-host relationships was considered disproportionate. Consequently, based on the applied correlation analysis we conclude that ARG abundance in these low-anthropogenic-impact datasets is likely not connected to individual but rather multiple hosts, allowing for subsequent analysis if overall community diversity is the hypothesized predictor of ARG abundance.

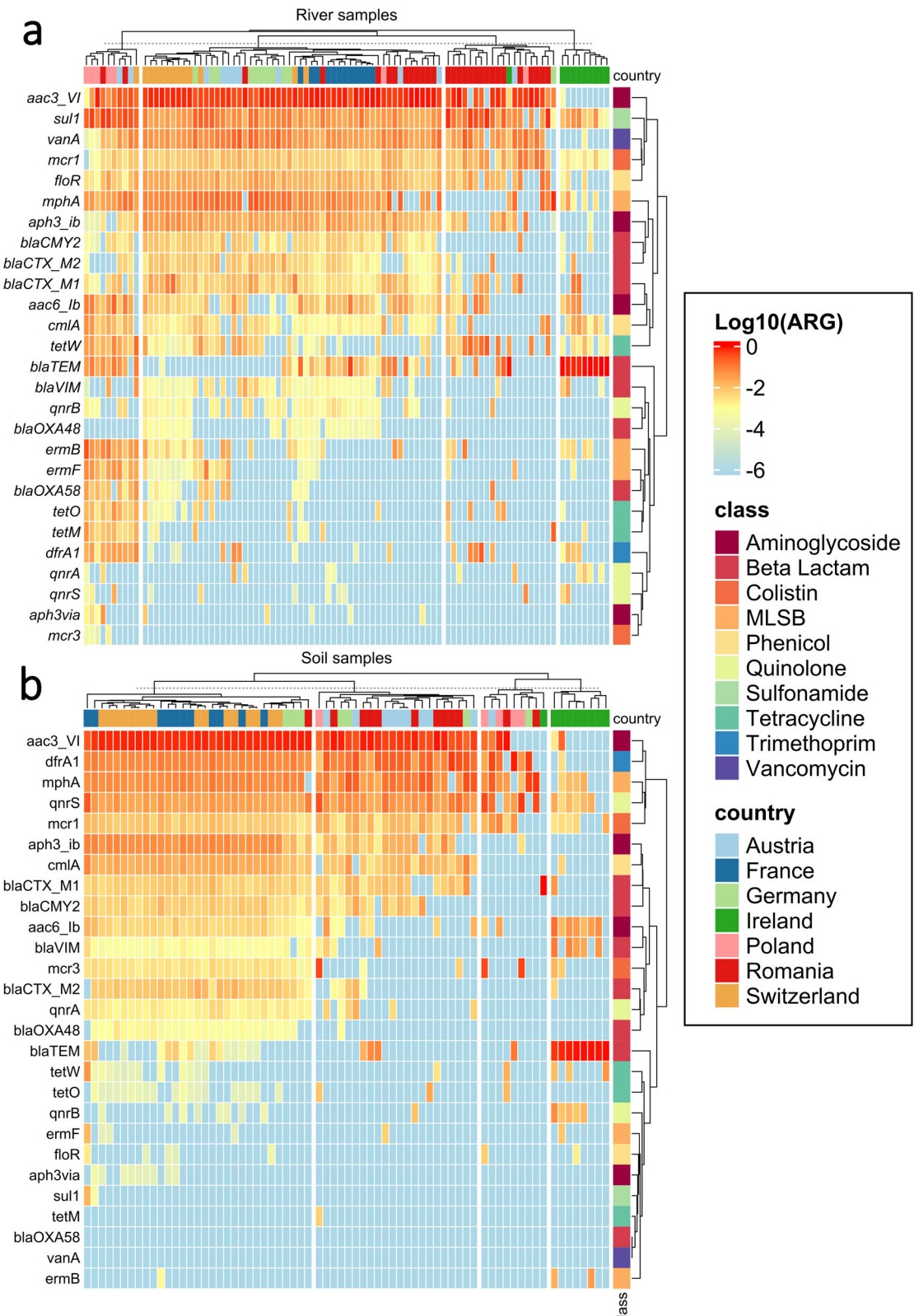

**Fig. 2 | Heatmap of relative ARG abundances. (a)** in the river dataset, **(b)** in the soil dataset. Values are displayed after transformation to log10 scale. The list of ARGs is presented based on similarity in abundance patterns and displayed from high abundance (red) to below the detection limit (blue). Color coding on the right displays the class of antibiotics they confer resistance to. Samples are ordered according to similarity in ARG profiles represented by the dendrogram on top based on default Euclidian clustering from the ComplexHeatmap R package[79] and color coded based on country of origin.

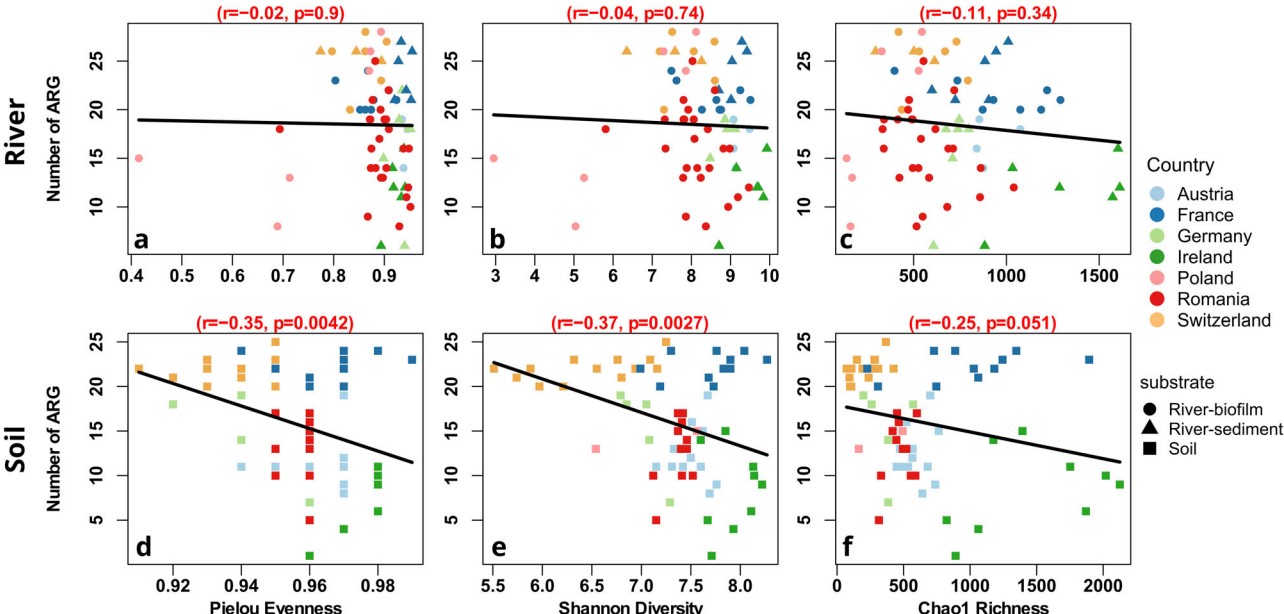

**Fig. 3 | Correlation analysis of the number of ARGs detected per sample with diversity metrics.** Linear Pearson correlation with Bonferroni correction for multiple testing from river environmental samples with Pielou Evenness (**a**), Shannon Diversity (**b**) and Chao1 Richness (**c**). Linear correlations from soil environmental samples with Pielou Evenness (**d**), Shannon Diversity (**e**) and Chao1 Richness (**f**). Colors depict the country of sample origin and the symbols depict the sample type.

## Diversity as a barrier to ARG spread

To determine whether higher community diversity could result in less diverse and lower abundant resistomes, we examined correlations between the diversity metrics and the number of detected ARGs as well as the relative abundance of each individual gene in both datasets. In the river dataset, no clear trend could be observed in correlation between the number of detected ARGs and any of the three diversity metrics. While all three correlations were negative, none were statistically significant ($r_{Pielou} = -0.02$, $r_{Shannon} = -0.04$, $r_{Chao1} = -0.11$, all $p > 0.05$; Spearman rank correlation; Fig. 3a-c).

Contrary to the river dataset, higher diversity in soils correlated with a lower number of detected ARGs. This negative correlation was significant based on Spearman rank correlation analysis for Pielou evenness ($r = -0.35$, $p = 0.0042$) and Shannon diversity ($r = -0.37$, $p = 0.0027$) (Fig. 3d, e). Similarly, for Chao1 Richness an inverse correlation with the number of ARGs detected was observed, however, barely not significant ($r = -0.25$, $p = 0.051$) (Fig. 3f). These results provided a first indication that diversity-based barrier effects might indeed exist, at least in the more structured soil environment. Although diversity had a significant impact, the effect sizes of 25-37% suggest that, as expected for complex environmental datasets, diversity is only one of multiple interacting drivers of the observed trends.

To further test this hypothesis the relative abundance of each individual ARG was correlated with the obtained diversity metrics. To account for zero-inflation during correlation, only those ARGs that were found in at least 25% of the samples of the respective dataset were tested. In the river dataset, similar to the number of ARGs no clear trends were observed for the relative abundance of any of the tested ARGs (Fig. 4a-c). The only correlation considered significant based on Spearman rank correlation (with Bonferroni correction for multiple testing) was a positive one between the $bla_{TEM}$ gene and Chao1 richness ($R_S = 0.42$, $p = 0.0003$). For the remaining combinations of ARGs and diversity metrics only slight negative or slight positive correlation trends ($R_S = -0.24 - 0.33$, all $p > 0.05$) were observed. Among those non-significant trends, no obvious patterns emerged. In fact, the average Spearman's rho of the tested ARGs for each of the three diversity metrics was near 0 ($p > 0.05$; t-test with Bonferroni correction for multiple testing).

In contrast, a high number of significant negative correlations of relative ARG abundance with the different diversity indices were observed based on Spearman Rank correlation analysis in the soil dataset (Fig. 4d-f).

Pielou evenness and Shannon diversity displayed the most significant correlations with 13 of the 18 tested ARG relative abundances being negatively correlated, while six ARGs were negatively correlated with Chao1 richness. ARGs negatively correlated with diversity were widely distributed across antibiotic classes. Similar to the river environment, the $bla_{TEM}$ gene was the main exception from the observed trend and positively correlated to either diversity metric ($R_S = 0.47$-$0.51$, all $p < 0.05$, Fig. 4d-f). Despite this outlier, a general trend for the correlation between relative ARG abundance and diversity was observed for the soil dataset, with the average correlation coefficients of all tested ARGs being both negative and significantly different from 0 for Chao1 richness ($R_S = -0.257 \pm 0.239$, $p = 0.0004$), Pielou evenness ($R_S = -0.234 \pm 0.228$, $p = 0.0001$) and Shannon diversity ($R_S = -0.267 \pm 0.223$, $p = 0.0003$, Fig. 4d-f).

Similar to ARGs, four of the five indicator genes for MGEs quantified in parallel through high-throughput qPCR (the class1 integron integrase gene $int$I1, the IncP plasmid $ori$T gene, the IncW plasmid $trw$AB gene, the orf37 of IS26) displayed negative correlation with all three diversity indices in the soil dataset (all $p > 0.05$), while no effect for the Tn5 transposase gene was observed (Fig. 5d-f). Again, in the river dataset, no or only slightly positive correlations of MGE abundance with the diversity indices were observed, mirroring the effects on ARG abundance (Fig. 5a-c). No significant correlation of the observed ARG number or the relative abundance of any ARGs with the relative abundance of crAssphage was obtained for the river dataset (all $p > 0.05$, Spearman, Supplementary Fig. 3) while crAssphage was entirely absent in all samples of the soil dataset. Thus, it again demonstrates that results are not directly impacted by recent anthropogenic fecal pollution.

## Similarity of communities according to total ARG abundance

Finally, we aimed at establishing if aside from community diversity, the abundance of ARGs in a sample is also predictable through phylogenetic similarity with samples of similar ARG contents. To achieve this, each of the two datasets was divided into three subsets: 1) the full dataset, 2) those 20% of samples with the highest total ARG abundance (Top 20%) and 3) those 20% of samples with the lowest total ARG abundance (Bottom 20%). Across both datasets, the samples of the Top 20% in total ARG abundance subset displayed significantly lower average pairwise Bray Curtis dissimilarity, and hence a higher degree of similarity, with each

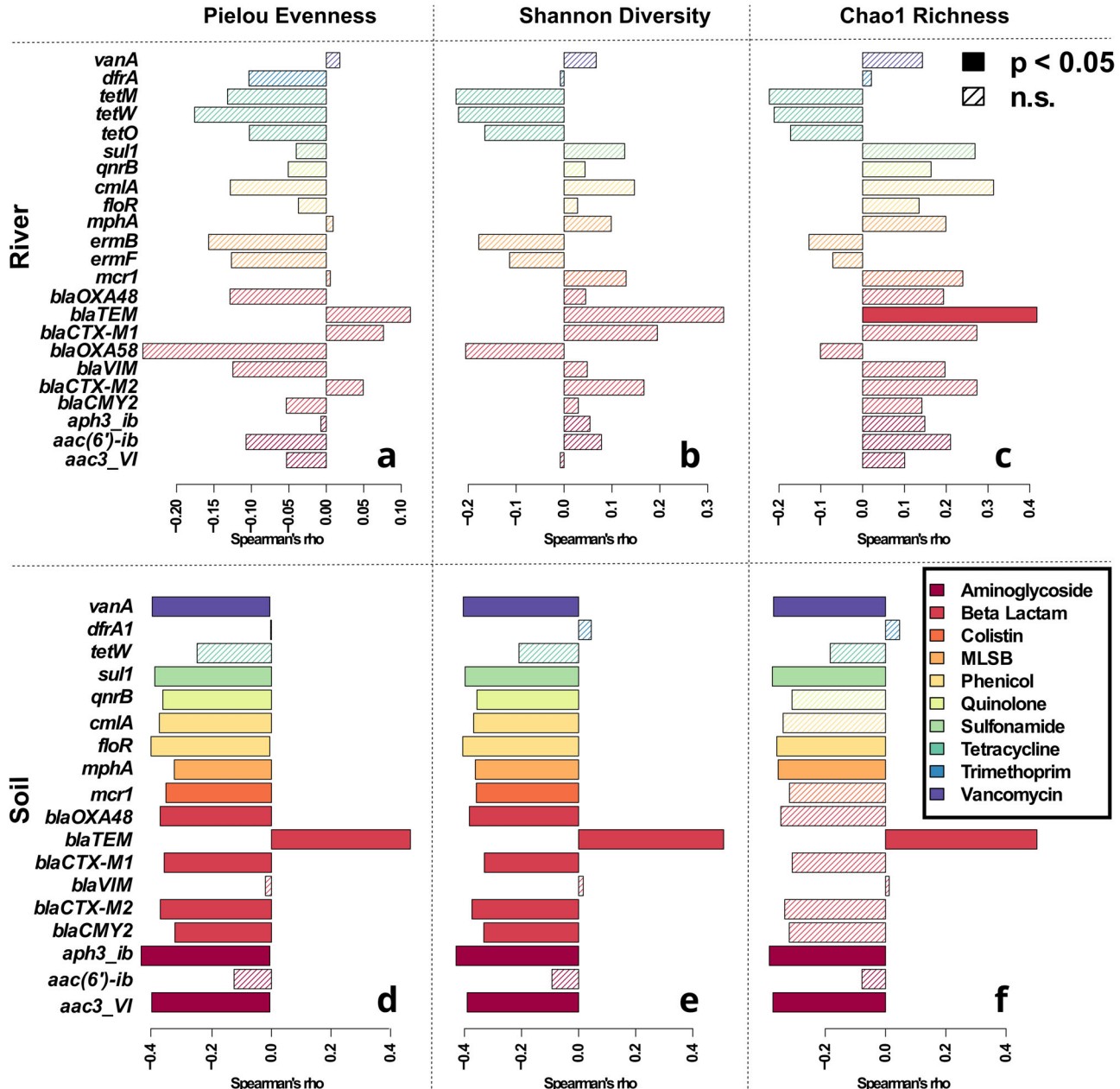

**Fig. 4 | Correlation analysis of relative ARG abundance with observed diversity metrics.** Spearman rank correlation with Bonferroni correction for multiple testing from river environmental samples with Pielou Evenness (**a**), Shannon Diversity (**b**) and Chao1 Richness (**c**). Correlations from soil environmental samples with Pielou Evenness (**d**), Shannon Diversity (**e**) and Chao1 Richness (**f**). Filled bars represent significant, while hatched bars represent non-significant correlations. Colors depict the class of antibiotic the ARG confers resistance to. Only ARGs that were detected in at least 25% of samples of a dataset were tested.

other compared to all samples in the full dataset (Fig. 6). In the river dataset dissimilarity decreased from 0.916 ± 0.089 (n = 4186 pairwise comparisons) in the full dataset to 0.873 ± 0.130 in the Top 20% of samples (n = 153) (p = 0.0066, one-way ANOVA with Tukey HSD). Similar in the soil dataset, dissimilarity decreased from 0.867 ± 0.109 (full dataset, n = 2016) to 0.821 ± 0.147 (Top 20%, n = 66) (p = 0.0199) (Fig. 6). Contrary, those 20% of samples with the lowest total ARG abundance displayed throughout a higher dissimilarity with each other than both the full sample dataset and the Top 20% dataset (all p < 0.05; Fig. 6). Consequently, while community diversity is correlated with ARG abundance in at least the structured soil environment, community similarity can only serve as a predictor for high ARG abundances, but is a bad predictor of low ARG abundance as low abundance samples have a high degree of dissimilarity.

## Discussion

Here we demonstrate based on analysis of a pan-European sampling campaign that communities of high bacterial diversity display lower diversity and abundance of ARGs, which provides indication that diversity might serve as a barrier to the long-term immigration and establishment of ARGs into environmental endemic microbiomes. Both, the number of detectable ARGs as well as a majority of the individual relative ARG abundances were negatively correlated with the diversity indices observed in soil. Among these indices, Pielou evenness was most significantly negatively correlated to the number of detected ARGs and their relative abundances. While this possible effect of community diversity on the pervasiveness of ARGs was highly visible and frequently statistically significant in the structured soil environment, it was barely observed in the river environment characterized by more frequent mixing events and bacterial community

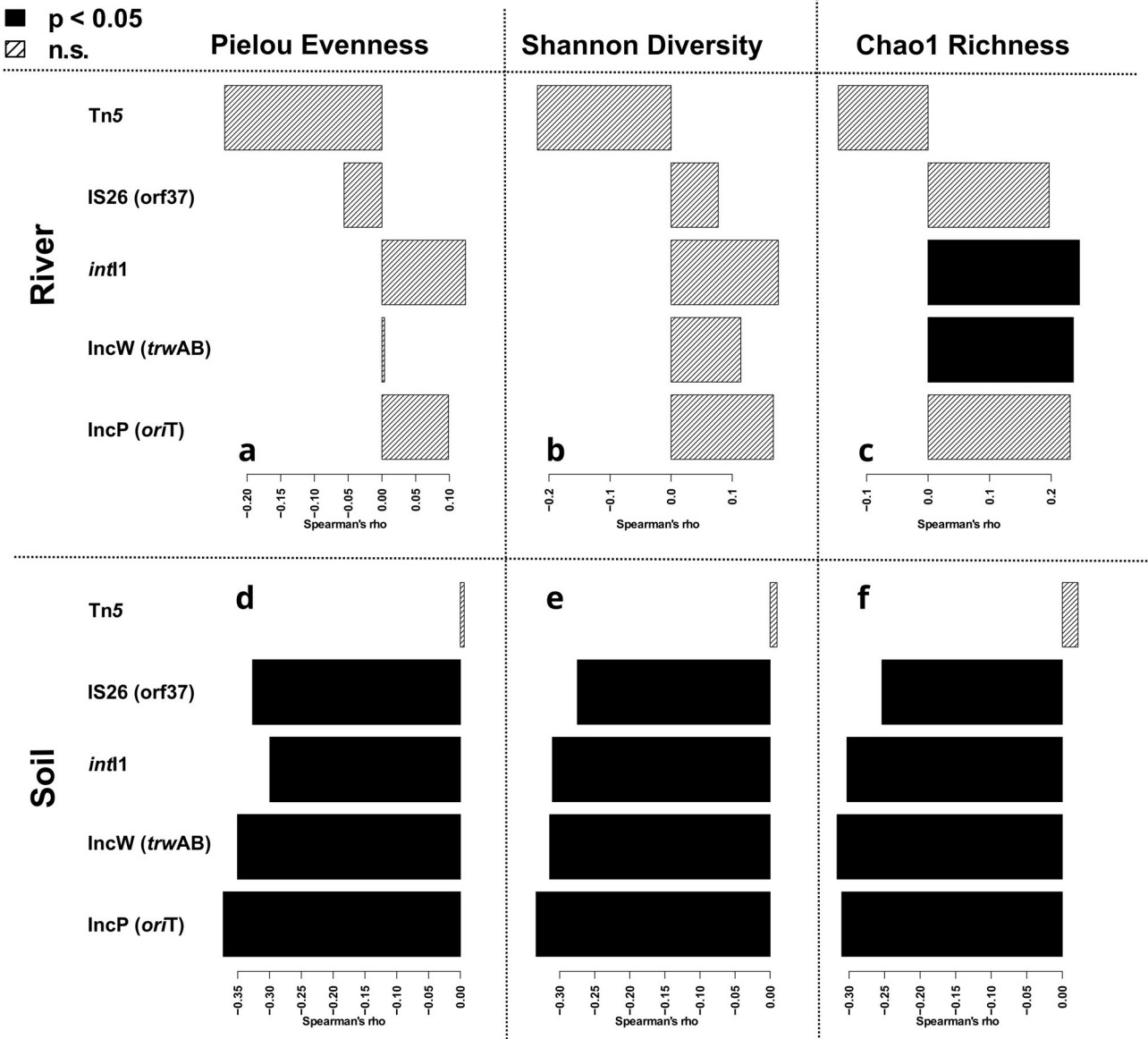

**Fig. 5 | Correlation analysis of relative MGE abundance with observed diversity metrics.** Spearman rank correlation with Bonferroni correction for multiple testing from river environmental samples with Pielou Evenness (**a**), Shannon Diversity (**b**) and Chao1 Richness (**c**). Correlations from soil environmental samples with Pielou Evenness (**d**), Shannon Diversity (**e**) and Chao1 Richness (**f**). Filled bars represent significant, while hatched bars represent non-significant correlations.

succession[35]. Crucially, the observed effects based on our statistical evaluation are consistent across the different analyzed ARGs, which serve as pseudo-replicates of individual genetic immigrants. Such a general effect across different ARGs would be expected if indeed general microbiome properties are the driving force underlying the observed ARG abundances. If, rather than a general diversity effect on ARG spread, other ecological drivers (e.g., the presence of chemical stressors) would be responsible, a more diverse effect would have been expected as the presence of such stressors would, while reducing overall diversity through inhibiting certain community members, particularly favor selection or co-selection of specific, individual ARGs.

Within the context of these results, the potential effects of diversity on invasion of AMR need to be assessed individually for the successive steps that make up a successful invasion event, namely 1) introduction of the invader, 2) its establishment, 3) its growth and spread along with 4) its impact in the new microbial community[18]. The initial introduction of an invader is primarily of stochastic nature and does not rely on biological interactions with the indigenous community[18,21]. Consequently, the success

of these initial introduction events depends on the quantity of invaders present, also known as the propagule pressure, together with the level of physical interaction of these invaders[22,36]. In this study, the samples originated from low impacted soils and rivers across Europe. Here, we define low impact as being not in direct proximity to the release of bacteria enriched in ARGs through anthropic action such as treated wastewater effluents[37,38] or manure[39,40]. The propagule pressure - the number of invaders harboring ARGs that were introduced into these environments - can be assumed low at the time of sampling and was likely low in the past. However, there is a high probability that bacteria with ARGs acquired in the antibiotic era occurred, nevertheless, at some rate (e.g., through human presence or transport by wild and domestic animals, including defecation, wet and dry atmospheric deposition), even if the exact rate of such rare invasion events over long timescales is impossible to determine. Consequently, it can be assumed that any increase in the resistomes in our samples are unlikely to stem from recent pollution events, but rather from past invasion events that manifest on top of the more or less universal background diversity and abundance of resistance recently determined for a number of environments[41,42]. Increases

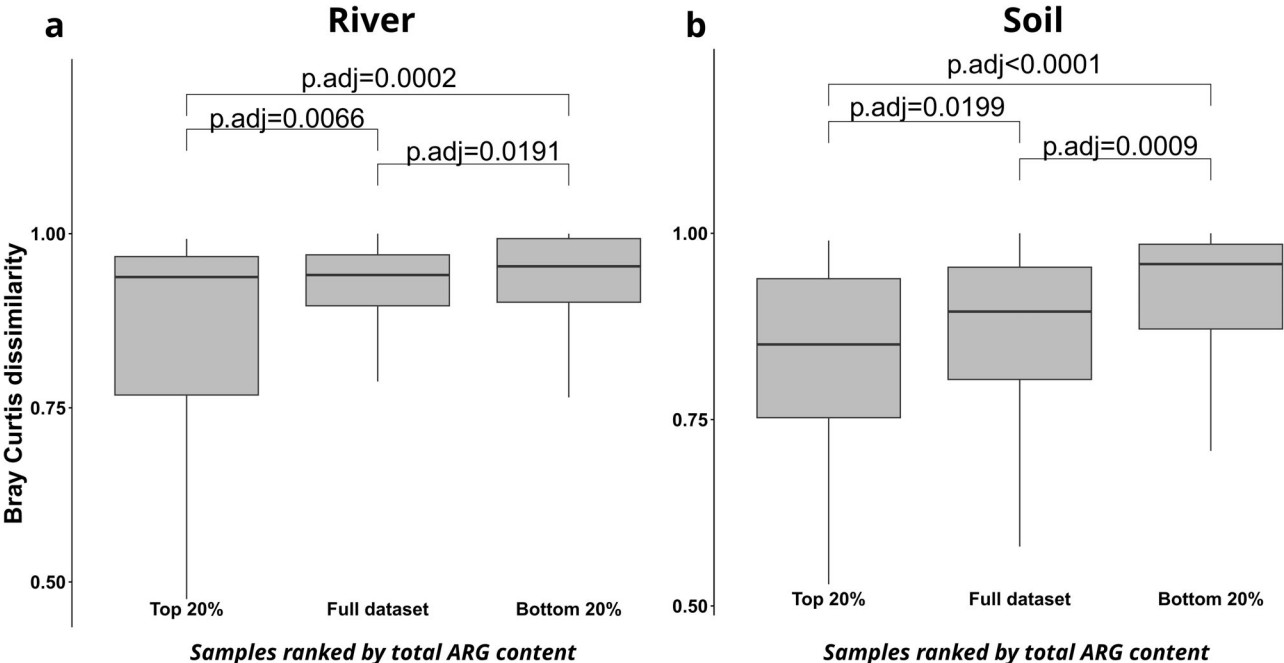

**Fig. 6 | Distribution of pairwise Bray-Curtis dissimilarities.** Distribution of pairwise Bray-Curtis dissimilarities for river (**a**) and soil (**b**) dataset is displayed across all samples within the dataset as well as across those subsets of samples with the 20% lowest (Bottom 20%) and the 20% highest (Top 20%) total ARG abundance. Significance testing between pairwise dissimilarity distributions is performed through one-way ANOVA with post-hoc Tukey HSD.

in ARG occurrence and relative abundance could hence result from the accumulation of invasion success of previous repetitive, but unquantifiable introductions of rare invaders over time that have been able to establish themselves in the autochthonous microbiome or left their mobile ARG load behind, if we consider that bacteria from the human or animal spheres are regularly not fully fit to be long-term maintained in environmental microbiomes.

Contrary to the original introduction step, where biological interactions play only a minor role, the interactions with the local community are highly relevant during the subsequent establishment and growth phases. The internal resistance of the indigenous community towards foreign microbes, e.g., its biotic barriers, have to be overcome to lead to the successful establishment of the invader[18,21] and can lead to the maintenance of ARGs in the community, when transient invasion success is long enough to allow for gene transfer[27]. Here we suggest that in the long term, bacterial diversity might provide a biotic barrier, hindering ARG success in the low impacted soil bacterial communities. However, no such effect could be observed for river communities. In the context of ARB such diversity effects have earlier been mainly demonstrated in short-term laboratory experiments for both types of environments using soil microcosms[29] and laboratory river flume experiments[24]. Still, in these experiments, diversity was artificially lowered to non-natural levels, which made it difficult to evaluate if such effects would equally be observable in the environment across natural biodiversity gradients in the long-term. In our analysis we make the implicit assumption that the present-day diversity is indicative also of past diversity, or at least of differences in diversity between sites and that both soil and riverbed microbiomes act as records of the long-term impacts endured by those microbial communities. This assumption appears likely to be correct in the case of forest soils, which are supposedly an environment typically stable over decades or more[43]. However, the assumption could be challenged regarding riverbeds, which are more likely exposed to considerably different conditions in the past (considering e.g., droughts, changes in water quality of European rivers over recent decades, etc.)[44]. This aspect may have contributed to the contrasting results in these environments.

The observed differences between the two datasets regarding the correlations between diversity and ARG abundances likely originate from the different nature of the two environments. The resistance of the community to the invasion processes is directly related to the number of available niches for invader establishment, with more diverse communities providing less available niche spaces, referred to in macro-ecology as the "diversity-invasion effect"[21]. In the stationary and structured soil environment, the amount of available niches rarely changes and is, once occupied by a diverse community, able to reach a steady-state[45]. Under these circumstances it is unlikely that niches open up for invaders as no major loss of community members is to be expected. Within this context it is also unsurprising that community evenness is more strongly correlated with lower ARG abundance in soil than community richness. In highly even communities, a higher number of bacterial species are abundant enough to fully occupy their specific available niche space[46]. Rich communities have an intrinsically increased potential for their populations to occupy a higher number of different niches[47]. However, once a certain degree of richness is reached, niche occupation moves towards saturation with additional species playing only a minor role[48]. Increased richness might then lead to increased competition and potential shifts within the metabolic networks, with novel niches becoming available and could be be exploited by invading strains. With these two mechanisms at play, the negative correlation of richness with relative ARG abundances remains weaker than that observed for evenness or Shannon diversity.

In the river microbiomes, bacterial diversity and available niches are potentially rather transient than long-term established due to the constant currents, biofilm adhesion and microbial dispersion events and alterations in nutrient availability[49]. Therefore, the effect of diversity on the long-term establishment of ARGs, which was observed for soil, may in the river microbiomes be masked by the dynamic nature in this environment. Moreover, the competitive ability of soil microbes against foreign bacteria might be elevated compared to those occupying riverbeds as several indigenous soil bacteria are known producers of antimicrobial compounds involved in competition and cell-to-cell signaling[50,51]. Less is known if such antimicrobial producers also occur in river biofilms at equal abundances, still any compounds produced would only transiently provide an advantage

before being washed away due to the rivers' aquatic nature. Hence, soil microbiomes can be thought of as more stable and more competitive in time than river environments. Still, soils are highly heterogenous in space[52], meaning that the strength of the here proposed barrier effect could also be highly variable on the spatial scale if niche occupation varies.

An interesting observation in this context was that the beta-lactam ARG $bla_{TEM}$ displays an opposing behavior to all other ARGs as its abundance was rather positively correlated with diversity in both datasets. In previous studies, $bla_{TEM}$ was regularly found to not correlate with other ARGs, crAssphage as an anthropogenic pollution indicator, or general introduction of solid or liquid human associated waste in both, soil[12,53] and aquatic[10,54] environments. Rather it was found at higher abundance in non-impacted environments as an integral part of the natural resistome[12,53,54]. Consequently, unlike for the majority of the other gene its main origin is likely not from anthropogenic enrichment and subsequent spread over long time periods.

Unlike for biotic invaders, such as alien bacteria, the invasion success of genetic invaders, such as ARGs, goes far beyond the successful establishment and growth of their hosts in the novel environment. A prolonged residence time of the host can favor the spread of mobile ARGs via horizontal gene transfer from the invading to the indigenous bacteria, thus leading to an even higher persistence of ARGs[27]. This can even be the case if the host's invasion is only transiently successful and invaders are lost after some time due to high community resilience in the observed environment. Mobile ARGs encoded on plasmids are able to spread from an invading donor strain to highly diverse proportions of soil and water derived bacterial communities, with bacteria belonging to over 25 different phyla able to receive individual resistance encoding plasmids[55–59]. However, effects of community diversity on the efficiency of horizontal gene transfer and the maintenance of plasmids in the community consist of a complex interplay of different mechanisms and remain difficult to predict. On the one hand, at higher diversity an increased number of potential plasmid hosts and conjugation partners are available that can lead to increased plasmid maintenance and transferability in the community[60,61] increasing the chance of transfer to a highly competitive host. On the other hand, in more diverse communities it can be harder to encounter a permissive conjugation partner, which reduces transferability due to this dilution effect[62]. Further, competition with other community members might increase the costs of resistance[63–65] and could ultimately drive the loss of ARG hosting plasmids from the community[66]. This loss process would be expected to be elevated in more diverse communities with better competitors. Our dataset provides a good indication, that it is rather the latter processes that are dominant in structured environmental communities as in our soil dataset, similar to ARGs a clear negative correlation for four of the five MGEs with diversity could be established. Hence, community diversity might also limit the horizontal acquisition of mobile ARGs from invading bacteria ultimately resulting in lower numbers and abundances of detected ARGs in the soil dataset. This is according to ecological theory, where species diversity is not always immediately implying a higher degree of genetic diversity[67,68]. Still, assuming that, in the long-term, invaders harboring the tested ARGs reach each of the tested communities it becomes apparent that an increasing number of ARGs are not successfully retained in those communities of higher diversity. If this is due to a shorter residence time of the invader, the above discussed increased competition, decreased horizontal gene transfer potential or dilution effects needs future research.

Finally, we tested if similarity in ARG abundances is predictable through phylogenetic similarity of the hosting microbiomes. Here, it became apparent that communities with high total ARG abundances are indeed phylogenetically closer to one another than would randomly be expected from the entire dataset. However, communities with low ARG abundances were even more dissimilar than the average samples dissimilarity in the entire dataset. This indicates that the ability to host ARGs at high abundances is rather a specialist trait and hence manifests in the phylogenetic composition of the community[69,70]. Contrary, low ARG abundances are not based on the phylogenetic composition of the community, but rather on the above demonstrated diversity-based barrier effects and can hence be characterized as a generalist community trait.

In summary, we display that the bacterial diversity within a given environment could affect the proliferation of AMR within and through this environment. Considering sites of low anthropogenic impact, the observed negative correlation of diversity with detection and abundance of ARGs in soils compared to river ecosystems can be directly connected to the intrinsic characteristics of the specific environments within the framework of invasion or immigration theory as well as horizontal gene transfer dynamics. Natural environments, such as rivers and soils, may play a key role in AMR development and proliferation. The characteristics of the individual environment, its dynamism as well as the diversity of the resident microbial community could define its role as a source or a barrier to AMR dissemination. We present support that in the structured soil environment, a high bacterial diversity might indeed serve as a barrier to the proliferation of ARGs in the autochthonous microbiome. Our results point to a previously overlooked benefit of healthy environments, with diverse microbial communities, providing natural barrier effects to the proliferation of AMR, thus clearly displaying how environmental and human health are immediately interconnected through the One Health concept. Furthermore, such barrier effects can be exploited within soil ecosystem management, for example, in defining optimal locations for aquifer recharge through wastewater reuse. Here choosing locations with a high intrinsic diversity could be beneficial in limiting the spread of wastewater born ARGs. To achieve this, the role of microbial diversity in the dissemination and mobilization of AMR markers requires a closer look through targeted experiments aimed at elucidating the exact mechanisms that limit the proliferation of resistance determinants and how exploiting such natural barrier effects could have cascading effects on the ecosystem biodiversity.

## Methods
### Soil sampling and processing
The terrestrial campaign consisted of collecting 74 forest soil samples from the seven countries during fall 2020 (Supplementary Fig. 4). The aim was to obtain sample sets that that are of relatively low anthropogenic impact (Supplementary Data 3).

From each forest location five single core samples (Pürckhauer drill, Buerkle™, Germany) were extracted from a depth of 0-25 cm along two 10 m virtual diagonals laid across the sampling location in the form of an X-pattern. 200 g of each of these five subsamples were combined in an aseptic plastic bag, thoroughly homogenized and transferred to the laboratory at 10 °C. From the composite sample aliquots of 20 g were sieved (2 mm mesh size) and stored at -20 °C. DNA extraction was performed using the DNeasy PowerSoil Pro Kit (Qiagen, Germany) according to the manufacturer's instructions. To obtain DNA from a total of 1 g of each sieved soil sample four replicates of 0.25 g each were extracted in parallel and combined thereafter. At least one extraction blank per country was used to confirm the absence of DNA contamination. The quality and quantity of the extracted DNA was assessed spectrophotometrically. Corresponding environmental metadata for the samples is, for those countries where it was available, included in Supplementary Data 3.

### Riverbed material sampling and processing
The aquatic campaign included the collection of 98 river samples from seven countries (Austria, France, Germany, Ireland, Poland, Romania, and Switzerland) during the fall/winter 2020/21 (Supplementary Fig. 4). The locations were selected to obtain samples that are of relatively low anthropogenic impact (e.g., no upstream wastewater treatment plant discharges; no known upstream discharge through agricultural activities or septic systems; no discharge through human recreational areas in the immediate proximity) (Supplementary Data 4). At each site, the substrate best representing sessile, non-phototrophic, oxygenated microbial communities in the chosen riverbeds, was identified through visual inspection and subsequently sampled. Either epilithic biofilms from the undersides of rocks to avoid phototrophic communities for those streams dominated by rock/

gravel, or oxygenated sediment for streams dominated by fine sediment were collected. Specifically, for epilithic biofilm samples, five individual rocks, collected from a shaded sample area from a riverbed length of approximately 10 meters, were gently scraped from the bottom surface using a sterile toothbrush, and combined to create a composite river biofilm sample. Repeated rinsing with sterile water in a 50 mL falcon tube was performed to collect the biomass. If no rocks or rock biofilms were available, fine surface sediment from shaded areas was sampled. In this case, the upper layer (~ 5 cm) of sediment was collected using a 50 mL falcon tube. Five sediment cores were combined at equal weight to obtain one composite sample.

All collected samples were gently homogenized and transported to the laboratory on ice. Then, samples were centrifuged (4,000 rpm for 5 min at 4 °C), the supernatant removed, pellets weighted and stored at −20 °C. DNA extraction was performed using the DNeasy PowerSoil Pro Kit (Qiagen, Germany) according to the manufacturer's instructions. At least one field blank of sterile water mixed with a sterile toothbrush used for obtaining of biofilms and one extraction blank per country were used to confirm the absence of DNA contamination. The quality and quantity of extracted DNA was assessed spectrophotometrically. Corresponding environmental metadata for the samples is, for those countries where it was available, included in Supplementary Data 4.

### Amplicon sequencing and analyses of sequence datasets

To analyze the bacterial diversity and taxonomic composition of the samples, DNA extracts were sent to the IKMB sequencing facility (minimum 10,000 reads per sample; Kiel University, Germany). Illumina MiSeq amplicon sequencing of the bacterial 16 S rRNA gene was performed using primers targeting the V3-V4 region (V3F: 5′-CCTACGGGAGGCAGCAG-3′ V4R: 5′-GGACTACHVGGGTWTCTAAT-3′)[71]. Sequences were collectively analyzed using DADA2[72] in QIIME2[73]. Forward and reverse reads were merged into amplicon-sequence variants (ASV), at 99% sequence similarity. Prior to downstream analyses, the *filter-features* and *filter-table* scripts were applied in QIIME2 to clean ASV and taxa tables by removing unclassified and rare ASVs with a frequency of less than 0.1% of the mean sample depth. The corresponding river and soil ASV tables consisted of 15,473 ASVs (river dataset) and 14,464 ASVs (soil dataset). All sequencing data was submitted to the NCBI sequencing read archive (SRA) under project accession number PRJNA948643.

### High-throughput qPCR of ARGs and genetic markers for MGEs

To determine the relative abundance of target genes in each sample, DNA extracts were sent to Resistomap Oy (Helsinki, Finland) for HT-qPCR analysis using a SmartChip Real-time PCR system (TaKaRa Bio, Japan). The target genes included 27 ARGs and 5 MGEs (Supplementary Data 5)[74]. In addition, the 16 S rRNA gene and the anthropogenic fecal pollution indicator crAssphage[32,33] were quantified. All samples were run with three technical replicates. The protocol was as follows: PCR reaction mixture (100 nL) was prepared using SmartChip TB Green Gene Expression Master Mix (TaKara Bio, Japan), nuclease-free PCR-grade water, 300 nM of each primer, and 2 ng/µL DNA template. After initial denaturation at 95 °C for 10 min, PCR comprised 40 cycles of 95 °C for 30 s and 60 °C for 30 s, followed by melting curve analysis for each primer set. A cycle threshold (CT) of 31 was selected as the detection limit[39,75]. The quantification limit was calculated as 25 gene copies per reaction accounting for 12.5 gene copies per ng of DNA template. Amplicons with non-specific melting curves or multiple peaks were excluded. The relative abundances of the detected gene to 16 S rRNA gene were estimated using the ΔCT method based on mean CTs of three technical replicates[76].

### Statistics and Reproducibility

To characterize the alpha diversity of the terrestrial and aquatic bacterial communities, the diversity indices Chao1 richness, Shannon diversity and Pielou evenness were calculated using the *core-metrics-phylogenetic* script in QIIME2[73] on rarefied data, with samples with an insufficient sampling depth

to reliably assess diversity removed from the datasets. The taxonomy of these bacterial communities was classified based on the SILVA classifier (version 138[77]). Beta-diversity dissimilarities among bacterial communities were assessed using a principal coordinate analysis (PCoA) based on Bray-Curtis distance matrices[78].

Differences in the relative ARG abundance for aquatic and terrestrial samples were transformed to a $log_{10}$ scale, and displayed using the package *ComplexHeatmap*[79]. To test the correlation between resistome and bacterial diversity, correlation analysis between relative ARG abundance and the calculated alpha diversity metrics was performed based on Spearman rank correlation or Pearson correlation followed by Bonferroni correction for multiple testing. Individual groups of data were compared using One-Way ANOVA with post-hoc Tukey HSD tests. Significant differences of correlations from 0 were tested using a t-test. All statistics and plots were produced using R version 4.2.0[80].

### Reporting summary

Further information on research design is available in the Nature Portfolio Reporting Summary linked to this article.

### Data availability

The datasets supporting the conclusions of this article are included within the article and its additional files. Original sequencing data is available in the NCBI sequencing read archive under project accession number PRJNA948643, with individual sample identifiers given in Supplementary Data 3 column AS & Supplementary Data 4 column AZ. Numerical source data for graphs is throughout included in the manuscript as Supplementary Data: Fig. 1—ASV tables underlying the creation of the PcoA plots is given in Supplementary Data 1 & 2, calculated alpha diversity indices based on ASV tables are given in Supplementary Data 3 column M-O & Supplementary Data 4 column S-U; Fig. 2—ARG abundances are given in Supplementary Data 3 column P-AO & Supplementary Data 4 column V-AU; Fig. 3 & 4—ARG abundances are given in Supplementary Data 3 column P-AO & Supplementary Data 4 column V-AU, while calculated alpha diversity indices based on ASV tables are given in Supplementary Data 3 column M-O & Supplementary Data 4 column S-U; Fig. 5—MGE abundances are given in Supplementary Data 3 column AP-AT & Supplementary Data 4 column AV-AZ, while calculated alpha diversity indices based on ASV tables are given in Supplementary Data 3 column M-O & Supplementary Data 4 column S-U; Fig. 6—Pairwise Bray-Curtis dissimilarities between samples based on ASV tables in Supplementary Data 1 & 2 are given as Supplementary data 6 & 7. Any additional information regarding the manuscript is available through the corresponding author upon reasonable request.

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

## Acknowledgements

We give a special thanks to Rosi Siber for creating the GIS Map. We thank all the local experts on soils and rivers in the different countries for advice regarding sample location identification. This work was supported by the ANTIVERSA project (BiodivERsa2018-A-452) here funded by the Bundesministerium für Bildung, und Forschung of Germany [01LC1904A], the French Agence Nationale de la Recherche [ANR-19-EBI3-0005-04], the Swiss National Science Foundation [186531], the Austrian Science Fund (FWF) [I 4374-B], the Irish Environmental Protection Agency [2019-NC-MS-9], the National Science Centre (NCN) of Poland [UMO-2019/32/Z/NZ8/00011], the Romanian National Authority for Scientific Research and Innovation (CCCDI – UEFISCDI) [117/2020]. UK, PF & TUB were supported through the Explore-AMR project funded by the Bundesministerium für Bildung, und Forschung under grant number 01DO2200. PF was supported through the China Scholarship Council (CSC) under grant number 202004910327. AT-M and ES were supported by the Ministry of Research, Innovation and Digitization through the Core Project BIORESGREEN, sub-project BioClimpact no. 7/30.12.2022, code 23020401. ES was partially supported by a grant from the Ministry of Research, Innovation and Digitization, CNCS - UEFISCDI, project number PN-III-P1-1.1-PD-2021-0529, within PNCDI III. DK & AXE were supported through the Urban Resistome project funded by the Deutsche Forschungsgemeinschaft (DFG) under project number 460816351. Responsibility for the information and views expressed in the manuscript lies entirely with the authors.

## Author contributions

Conceptualization of the study and sampling strategy: UK, GG, EC, XB, SG, AGS, ES, CC, NK, MP, FW, MW, HB, CM, TUB; Identification of national sampling locations, sampling, metadata collection & sample processing: UK, GG, EC, XB, ID, SG, AGS, UO, ER, MS, ES, AT-M, CC, NK, MP, JV, FW, MW, HB, CM, TUB; Sampling coordination, sequencing and HT-qPCR data acquisition: UK, TUB; 16 S sequence analysis: GG, EC, AXE; HT-qPCR analysis: UK; Data curation and validation: UK, GG, EC; Correlation and network analyses: UK, GG, EC, DK, PF, AXE; Data interpretation: UK, GG, EC, HB, CM, TUB; Visualization of data: UK, GG, EC, DK, PF; Funding acquisition: CC, NK, MP, FW, MW, HB, CM, TUB; Supervision: UK, CC, NK, MP, JV, FW, MW, HB, CM, TUB; Writing - original draft: UK, GG, EC; Writing - review and editing: all authors. All authors have read and approved the final version of the manuscript.

## Funding

## Competing interests

The authors declare no competing interests.

## Additional information

[1]Technische Universität Dresden, Institute for Hydrobiology, Dresden, Germany. [2]Eawag, Swiss Federal Institute of Aquatic Science and Technology, Department of Surface Waters – Research and Management, Kastanienbaum, Switzerland. [3]Université de Lorraine, Villers-lès-Nancy, France. [4]Université de Toulon, Toulon, France. [5]TU Wien, Institute of Water Quality and Resource Management, Vienna, Austria. [6]AGES – Austrian Agency for Health and Food Safety, Department for Integrative Risk Assessment, Division for Risk Assessment, Data and Statistics, Vienna, Austria. [7]University of Warsaw, Faculty of Biology, Institute of Microbiology, Department of Bacterial Physiology, Warsaw, Poland. [8]Warsaw University of Life Sciences, Institute of Biology, Department of Biochemistry and Microbiology, Warsaw, Poland. [9]Maynooth University, Department of Biology, Kathleen Lonsdale Institute for Human Health, Maynooth, Co. Kildare, Ireland. [10]Romanian Academy of Science, Institute of Virology Stefan S. Nicolau, Bucharest, Romania. [11]NIRDBS, Institute of Biological Research Cluj-Napoca, Cluj-Napoca, Romania. [12]Interuniversity Cooperation Centre Water & Health, Vienna, Austria. [13]These authors contributed equally: Uli Klümper, Giulia Gionchetta, Elisa Catão. [14]These authors jointly supervised this work: Helmut Bürgmann, Christophe Merlin, Thomas Ulrich Berendonk.
✉e-mail: thomas.berendonk@tu-dresden.de

