## [Peer Review File · Communications Biology]

Reviewers' comments:

Reviewer #1 (Remarks to the Author):

A well sized study with appropriate use of amplicon sequencing and high throughput qPCR to provide new insight into environmental AMR across geographies and matrices. The sites studied focus on locations with little anthropogenic impact (which could be a bit better substantiated), which is less studied as much focus is on supposed 'hot spots' of resistance with large urban or agricultural impacts. A major limitation of the present study is that data is only available from the DNA analyses. The ability to compare the observations presented and the environments studied is severely limited for the soils by not knowing basics of the soil properties (e.g., TOC, sand-silt-clay, pH, conductivity) and similarly for the river samples as no water quality/sediment quality/biofilm (e.g., EPS for the latter) parameters are reported. It is not clear without the aforementioned quality measurements how the team selected samples 'to capture varying diversity.' It is not clear to the reviewer that a cross-matrix study of environments without documentable ARG/ARB inputs is the correct design for addressing immigration of ARBs and ARGs. An observational field study without treatment (i.e., introduction of new microbes) or measurement of inputs (e.g., here atmospheric deposition?) does not seem to be proper design for measuring immigration/proliferation because it is not clear that there was an opportunity for immigration in these lightly impacted environments.

A more minor suggestion is that terminology in several places could be more precise – for example, the term "level" is used for ARGs or ARBs would better expressed via commonly quantified metrics (concentration, diversity, expression, etc.).

Comments:

First sentence of the abstract is difficult to follow – does this mean the focus is on whether immigrating ARBs/ARGs persist/replicate?

L73: lower levels means lower concentrations? Less diversity? Less severe ARGs?

L65: ARGs are natural, Line 74: ARGs of anthropogenic origin...with exact sequence similarity between each, are these statements not contradictory? Is "anthropogenic introduction" or a more precise phrasing possible

L97: For resilience, what is the metric?

L85: provide an example of a small-scale disturbance

L105-6: But the argument is also made that abundant resources and proximity promote HGT and AMR proliferation in activated sludge communities

L114: the authors above suggest atmospheric deposition, is this to mean no point and nonpoint source of solid or liquid waste discharges?

Methods

L426-7/442-43: How could one know a priori that samples would have a range of diversities? Was soil/sediment type (i.e., organic matter, sand-silt-clay, conductivity, etc. controlled for?) These parameters do not appear to have been measured and themselves are major drivers of diversity, no?

L442/443: "not exposed to WWTP" -> "no upstream wastewater treatment plant discharges". Confirm no upstream agricultural activity, septic systems. Also note the proximity to recreational areas with latrines or human recreational use areas

Table S2: Provide estimates of stream flow volume and/or widths/depths be provided

L1445: O2 was measured? Or based on other field observations of mixing/lack of methane odors?

L449: shadowed = shaded?

Please describe field blanks and frequency

L487: specify quantification limit, describe calibration curve (efficiency, R2). presumably all samples run in triplicate (technical replicate)?

L145-6: Would this be expected? Geographic borders may not coincide with differences in important environmental parameters such as river flow rate or forest type or soil type or temperature or source water (e.g., snow melt vs groundwater...can no other drivers/factors be explored in these data sets?

Results/Discussion

Section beginning L163 – please correct ARG notation – convention not all italics for gene names, only first 3 letters

Section beginning L195: The significance of these comparisons is not clear to the reviewer.

L228-262: While 27 ARGs is an excellent panel for PCR, there are hundreds detected via metagenomics, so testing correlations of total presence/absence of this subset is not fully justified. The fact that these correlations are generally not maintained when testing an individual ARG also seems to undermine the hypothesis. Were these correlations also tested with the crAssphage marker?

L287: Team demonstrates high diversity has less presence of the 27 ARGs tested, but “resistance to pervasiveness” is not clear and the latter part of this statement is one speculation as to why this was seen

L296: The repetition of the “more dynamic river environment” needs justification – observations here were for single sampling events

Paragraph L297-399: Quite speculative – no events where invasion could occur were documented...this information can be included in abbreviated form in the discussion, but mostly the study design here isn't appropriate to answer questions about invasion as no invasion events were documented. This is a missed opportunity to instead a comparison to other pristine environments and ecological drivers

Reviewer #2 (Remarks to the Author):

Summary and general comments:

The manuscript by Klümper et al. uses a unique suite of soil and river sediment & biofilm samples from across Europe to address their hypothesis that bacterial diversity helps inhibit the ‘invasion’ of antibiotic resistance genes (ARG) into the environment, with implications for the propagation of anthropogenically-sourced ARG inputs via wastewater effluents or application of manure in agricultural systems. The authors utilize the aforementioned set of soil and river samples that span bacterial diversities (via 16S rRNA gene sequencing) to both compare systems that vary in stability and identify intrasystem influences of diversity on ARGs (identified via qPCR). Overall, I believe that the data are solid and that this is an impactful contribution to the research community for assessing future ecosystem risk to anthropogenic ARG inputs – and such a cool story with a neat dataset! However, I think that there are some improvements that should be made to refine the manuscript and further validate the findings & explore the dataset for final publication.

Major comments:

While the data is comprehensive and is generally overviewed well within the text, I think there is some data and results that are not discussed as thoroughly as needed. I want to note that this dataset is super cool and unique & there is a LOT you could do with it – so I try to only point out things here that I believe would add to the story. One such piece of data was the 16S rRNA gene sequencing data, which was only used to assess diversity. It would be interesting to see whether there are any specific taxa that were any taxa-ARG correlations – i.e., are any bacterial taxa affiliated with the enrichment of a given ARG? Did soil or river samples with a higher abundance of ARGs harbor similar bacterial communities? You use the term ARB – antibiotic resistant bacteria – but then do not discuss any specific taxa that are correlated with ARGs & only briefly mention the phyla-level differences between the soil & rivers, which is already fairly well-characterized. There is also some language in the introduction regarding ‘invaders’ and how, if they are phylogenetically close to the established community, they will have greater success (L82), but again none of the analyses focus on ARBs.

Another piece of data that I feel wasn’t discussed enough was the abundances of blaTEM and how different its profile was compared to other surveyed genes (e.g., consistently high abundance in Ireland sites and it being the only gene with a significant positive correlation to diversity in both systems. This clear distinction between it and the other ARGs kept popping up to me as a reader and it needs to be discussed more than just pointing it out – is there any reason why this ARG would act so differently to the others? It especially needs to be discussed because it is consistently so different. Further, I think there could be more discussion on specific ARGs, like why is there not a significant correlation with some of them (e.g., dfrA1, aac(6′)_ib, etc.)?

Something else that was briefly mentioned but not discussed as thoroughly as it should have been was the mobile genetic element (MGE) data. You first mention them in L275, then briefly discuss them in the discussion (~L390). I also see them in Table S3, but do not see any of the actual abundance data from them in Table S1 or S2. If this data is going to be included in the manuscript, the data needs to be included in Table S1 and S2 and you need to include a SOM figure showing the lack of correlation between MGEs and diversity. This is another cool piece of data that adds to the story but I don’t feel is utilized enough.

Along with these pieces of data that I feel could be more thoroughly discussed or intertwined throughout the manuscript, I did not feel like that ‘Higher degree of correlation between relative ARG abundances in soils compared to rivers’, along with Figure 3, added much to the story. I just didn’t fully follow how the correlation between specific ARG abundances fit or informed the hypothesis. I may be completely missing something (& please push back if so!) & this language needs to be tightened and made clearer, or this section could be shortened to make room for some of the other analyses I suggested above.

Lastly, is there any chemistry metadata on any of these samples, specifically the soil samples? It seems odd to have all of these samples and only the microbial data, when we know that in environmental systems the chemistry and microbiology have complex feedbacks, which likely also influence ARGs. This might be out of scope of this study & might be data that’s being used somewhere else but, as a soil microbiologist, we rarely publish soil microbiome data without including chemistry metadata.

Minor comments:

L92 – ‘as this usually coincides with a lower rate of niche occupation’, this needs a reference. I know you cite references that would work here above, but something needs to be here. “Usually coincides..” is too arm-wavy of a statement to not be followed by a citation that supports

L99 – L108 – Just want to say that I love this paragraph! It frames the study nicely.

L112 – The use of ‘long-term’ should be removed here, it suggests a time element to this study.

L122 & L125 – Examples of inconsistent tense used here (e.g., ‘while the other half was obtained’ should be ‘while the other half were obtained’). Keep tense consistent throughout text.

L124 – Introduce the term resistome here (e.g., ‘microbial diversity was assessed through 16S rRNA gene-based amplicon sequence, while the resistome was analyzed via the abundance of 27...’). This term is just dropped in the results section but never explained and some readers might not be familiar.

L124 – ‘bacterial 16S rRNA gene-based amplicon sequencing’ should be changed to just ‘16S rRNA gene sequencing’. The fact that its 16S implies that its bacterial, and ‘microbial diversity’ should be changed to ‘bacterial diversity’ because microbial could include fungi & viruses along with bacteria.

L127 – I recommend moving the hypothesis statement to the end of this paragraph to leave the reader clearly knowing before going into results.

L131 – Because the methods are at the end in this journal format, I’d make sure to nod towards the methods throughout the results section. E.g., here it is unclear what the ‘river’ samples are. Be clear that these are not surface water samples, but are river sediment & biofilm samples. The use of the term river is very vague.

L132 – Again, instead of ‘microbial communities’ I’d use ‘bacterial communities’ so as not to imply the inclusion of fungi or viruses. Folks especially think about fungi in soils and could misinterpret the term as inclusive of fungi.

L134 – L135 – Why are these n values so much larger than the total number of samples from both systems that are mentioned in the first sentence of this paragraph? I don’t understand how you have 148 river biofilm samples when you mention 94 in L131. Did this mean 94 rivers or samples? Be very clear here, as the methods aren’t until the end of the manuscript and you don’t want readers to have to search.

Fig 1A – There’s a lot going on here, I’d add some ellipses around the samples representing different substrates to help the reader see the differences between river & soil samples that you mention in the text.

L137 – The explanation that some river communities differed between country seems misplaced here

and does not add to the story. I recommend removing this note.

L143 & L144 – Change the values to percents (I think they are currently relative abundance out of 1?). You mention relative abundance previously (L141) so should keep it consistent.

L143 – Minor but Figure S2 is mentioned before Figure S1, should renumber so that they are in order.

L145 – Whenever you use term ‘significant’ should be along with some sort of statistical test. Did you do a stat test here? If not, remove.

L146 – Need a reference with the introduction of CrAssphage.

Fig 1B – Would be helpful if the river and soil values were plotted next to one another instead of above & below, especially because you directly mention the difference between them.

L163 – In this title, you mention diversity but don’t really discuss diversity of the ARGs. You do a bit later with noting the # of ARGs in different samples, but I think it would be helpful here to include a SOM fig that shows diversity of ARGs between samples.

L164 – Briefly state the methods here before jumping into the data. E.g., ‘We analyzed the resistome of our samples using qPCR of X different genes...’.

L176 – Minor but I’d order these as the figure is ordered. Also, acc(3)-VI needs to be aac(3)-VI.

Figure 2 – Edit the y axes so that they are the polished gene names (like what you have in Fig 5) and increase text size of legend.

Figure 3 – Again, gene names should match how they are written in Fig 5. Also difficult to read names here. Need to add label legend.

L208 - 211 – This sentence reads confusing to me and is maybe where I got mixed up on the significance of this section and these findings. Please tighten language and relevance of the findings to the hypothesis. Further, state here what kind of network analysis was used (WGCNA?). The reader should have to go to the methods for specific details but not just the type of analysis!

L212 – Could you show this in a SOM figure?

L227 - 228 – The statement about the number of ARGs detected in samples in the river (these lines) and soil (L241 – L242) seems out of place here and should be at the beginning of the resistome diversity and abundance section.

Figure 4 – This figure should be plotted the opposite way – the diversity value should be on the x axis and number of ARGs on the y axis because you’re implying that the number of ARGs is the dependent variable here.

Figure 5 – I really, really love this figure! Elegant way to show a lot of data & tests at once. Make the legend and axes text a bit bigger.

L275 – Vague here what ‘tested’ means – do you mean targeted via qPCR? Further, this is already mentioned in the major comments section, but the MGE data should be elaborated on! Plus, lack of correlation would be nice to see in SOM fig.

L407 – The use of ‘texture’ here is odd and only applies to soil & river sediment, not so much the river biofilms. Plus, in the discussion you explain some of the dynamic nature of rivers (L361-363) but don’t talk about soils much. While soils can be thought of as more stable in time than river environments, I think it would help to give a nod to the fact that soils are very heterogenous in space as compared to time and how this may influence your findings or the implications of this on the proliferation of ARGs in soils.

L439/sampling details – You should be very clear throughout the text that the river samples are sediment and biofilms, not surface water. As an environmental micro person, I was a bit confused at the beginning of the manuscript what ‘river samples’ entailed. Make sure you’re very specific throughout the entirety of the text so the reader knows what environment you’re discussing.

We thank the editor and the two reviewers for their time, dedication and helpful assessment of our manuscript. We believe that based on the reviewers' inputs and critical assessment the manuscript has now substantially improved compared to the earlier version. Please find below a detailed response to each individual reviewer comment (numbered to allow cross-referencing). Responses are throughout given in blue, with relevant changes in the revised manuscript cited in *blue italics*. Additionally line numbers referring to the new clean version of the manuscript are referenced for easy checking of any edits, additions, etc..

Responses to Reviewer #1:

R1.1: A well sized study with appropriate use of amplicon sequencing and high throughput qPCR to provide new insight into environmental AMR across geographies and matrices. The sites studied focus on locations with little anthropogenic impact (which could be a bit better substantiated), which is less studied as much focus is on supposed 'hot spots' of resistance with large urban or agricultural impacts.

Response: Thank you for your positive, critical and helpful comments.

R1.2: A major limitation of the present study is that data is only available from the DNA analyses. The ability to compare the observations presented and the environments studied is severely limited for the soils by not knowing basics of the soil properties (e.g., TOC, sand-silt-clay, pH, conductivity) and similarly for the river samples as no water quality/sediment quality/biofilm (e.g., EPS for the latter) parameters are reported. It is not clear without the aforementioned quality measurements how the team selected samples 'to capture varying diversity.'

Response: The reviewer is right and we are aware of this. The mentioned limitation is based on the origin story of the dataset: The sampling campaign was originally carried out to screen a large number of soils and river biofilm/sediment samples in order to identify suitable soils/ivers of differing diversities for short-term laboratory invasion experiments with resistant bacteria (e.g., <https://doi.org/10.1016/j.scitotenv.2023.166661>). Hence, only a limited amount of metadata was collected as sample collection was carried out in exploratory manner in hopes of gaining a high variety of diversity levels rather than actively pursuing it through a priori metadata evaluation. A main reason for this was the very differently equipped project partners in the 7 different countries with expertise and specialization on different matrices.

However, once we realized that the captured diversity in the dataset was indeed highly variable, we were wondering if diversity would not only affect the short-term invasion of resistant bacteria (as mentioned above), but also affect long-term levels of resistance in the community, which we address through the analysis presented in this manuscript with high effect sizes for the soil samples.

To further address this comment, we have now collected as much metadata as was available for the samples based on previous or retrospectively possible analysis of samples available through the different partners and added it to the SI tables. Still, the available metadata remains largely heterogenous across sample type and countries. While we now tested if any of the soil or water properties significantly affected ARG levels, no significant patterns emerged. This could be either due to them not existing or due to the lack of data and hence statistical power for certain parameters for a large proportion of samples. As this additional analysis was not very informative due to the restrictions mentioned above, we have hence decided to not include it in the revised manuscript, but rather to only provide the available metadata in the SI tables.

R1.3: It is not clear to the reviewer that a cross-matrix study of environments without documentable ARG/ARB inputs is the correct design for addressing immigration of ARBs and ARGs. An observational field study without treatment (i.e., introduction of new microbes) or measurement of inputs (e.g., here atmospheric deposition?) does not seem to be proper design for measuring immigration/proliferation because it is not clear that there was an opportunity for immigration in these lightly impacted environments.

Response: The reviewer is right, that we do not artificially introduce new microbes, nor have a definite measure of the inputs (e.g., atmospheric deposition etc.) in our study. Such would definitely be necessary when addressing short- or mid-term immigration dynamics into environmental communities. We and others have indeed already addressed such short-term invasion dynamics of single introduced strains or communities (anthropogenically enriched in AMR) into environmental communities of different diversity in other manuscripts (e.g. Chen *et al.* 2019; Bagra *et al.* 2023, 2024; Gionchetta *et al.* 2023) and equally demonstrated a diversity-based barrier effect. However, such controlled experiments are limited by the fact that, in order to detect effects, usually large amounts of invaders are necessary, which cause severe disturbances of the community, which is not a realistic scenario in low-impacted environments.

In such low-impacted environments the true relevance of such low-level immigration events is furthermore not the short-term impact which would be measurable in direct invasion experiments, but if such immigrant bacteria and genes upon their rare arrival are able to persist over long, evolutionary timescales and leave a mark in the natural communities.

While we agree that it would be optimal to have exact, quantifiable inputs in our cross-matrix environmental field-based study over time, obtaining such data for natural environments over long ecological/evolutionary timescales is impossible, making a comparative field-based study the only way of investigating such processes in the proper natural context. One strong indication that this is indeed an appropriate way of testing the set-out hypothesis is that the effects are consistently found across different ARGs that can be seen as “pseudo replicate invading genes”. If, rather than a diversity effect on their immigration, other ecological drivers (e.g., conditions favouring selection for specific ARGs) would be responsible, we would expect to see a more ARG specific rather than a general response on all ARGs. This notion is now also represented in the discussion section (L. 346-352):

“Crucially, the observed effects are based on our statistical evaluation consistent across the different analyzed ARGs, which serve as pseudo-replicates of individual genetic immigrants. Such a general effect across different ARGs would be expected if indeed general microbiome properties are the driving force underlying the observed ARG abundances. If, rather than a general diversity effect on ARG immigration, other ecological drivers (e.g., the presence of chemicals favoring selection for specific ARGs) would be responsible, a more diverse, ARG-specific effect would have been expected.”

To further address the reviewers comment we now clarified the fact that we are not able to quantify any of the inputs in the discussion section, but we still strongly believe that they occur rarely but regularly over evolutionary timescales (L.365-377):

“However, there is a high probability that bacteria with ARGs acquired in the antibiotic era occurred, nevertheless, at some rate (e.g., through human presence or transport by wild and domestic animals, including defecation, wet and dry atmospheric deposition), even if the exact rate of such rare invasion

events over long timescales is impossible to determine. Consequently, it can be assumed that any increase in the resistomes in our samples are unlikely to stem from recent pollution events, but rather from past invasion events that manifest on top of the more or less universal background diversity and abundance of resistance recently determined for a number of environments^{41,42}. Increases in ARG occurrence and relative abundance would hence result from the accumulation of invasion success of previous repetitive, but unquantifiable introductions of rare invaders over time that have been able to establish themselves in the autochthonous microbiome or left their mobile ARG load behind, if we consider that bacteria from the human or animal spheres are regularly not fully fit to be long-term maintained in environmental microbiomes.”

References used in the answer above:

Bagra K, Bellanger X, Merlin C et al. Environmental stress increases the invasion success of antimicrobial resistant bacteria in river microbial communities. *Sci Total Environ* 2023;904:166661.

Bagra K, Kneis D, Padfield D et al. Contrary effects of increasing temperatures on the spread of antimicrobial resistance in river biofilms. McMahon K (ed.). *mSphere* 2024;9, DOI: 10.1128/msphere.00573-23.

Chen QL, An XL, Zheng BX et al. Loss of soil microbial diversity exacerbates spread of antibiotic resistance. *Soil Ecol Lett* 2019 11 2019;1:3–13.

Gionchetta G, Snead D, Semerad S et al. Dynamics of antibiotic resistance markers and *Escherichia coli* invasion in riverine heterotrophic biofilms facing increasing heat and flow stagnation. *Sci Total Environ* 2023;893:164658.

R1.4: A more minor suggestion is that terminology in several places could be more precise – for example, the term “level” is used for ARGs or ARBs would better expressed via commonly quantified metrics (concentration, diversity, expression, etc.).

Response: We have carefully gone through the manuscript to improve the terminology throughout and increase the precision. For example, where possible, level, when referring to ARGs/ARBs was replaced through either diversity or abundance.

R1.5: First sentence of the abstract is difficult to follow – does this mean the focus is on whether immigrating ARBs/ARGs persist/replicate?

Response: We have rewritten the first sentence of the abstract to clarify the focus, which is not in persistence and replication, which are rather short-term processes we tested in separate studies (e.g. <https://doi.org/10.1016/j.scitotenv.2023.166661>; <https://doi.org/10.1128/msphere.00573-23>), but rather immigration over time and becoming part of the indigenous microbiome in the long-term (L. 40-42):

“When antimicrobial resistant bacteria (ARB) and genes (ARGs) reach novel habitats, they can become part of the habitat’s microbiome in the long term if they are able to overcome the habitat’s biotic resilience towards immigration.”

R1.6: L73: lower levels means lower concentrations? Less diversity? Less severe ARGs?

Response: It means lower diversity and abundance and has now been replaced accordingly.

R1.7: L65: ARGs are natural, Line 74: ARGs of anthropogenic origin...with exact sequence similarity between each, are these statements not contradictory? Is “anthropogenic introduction” or a more precise phrasing possible

Response: We have now rephrased this sentence to clarify that it is dealing with *“ARGs or ARBs enriched and introduced through anthropogenic action”* (L. 75).

R1.8: L97: For resilience, what is the metric?

Response: We have now added an explanation to this sentence (L.99-101):

“Alternatively, future invasion events could be favored by reducing the community resilience, through, for example, reduction of microbial network connectivity and hence the competitiveness of the indigenous community”

R1.9: L85: provide an example of a small-scale disturbance

Response: We now provide an example for a small-scale disturbance (L86-87):

“In contrast, even small-scale disturbance events, such as exposure to a novel stressor, can considerably increase invasion success by affecting the niche occupancy of resident species.”

R1.10: L105-6: But the argument is also made that abundant resources and proximity promote HGT and AMR proliferation in activated sludge communities

Response: We have now incorporated this notion into the manuscript (L. 108-113):

“Further, diverse microbial communities with a high degree of functional niche coverage, such as activated sludge, have been suggested to provide natural barriers for the proliferation of AMR. However, data suggests that activated sludge communities have also incorporated a particularly high diversity of ARGs encoded on mobile genetic elements (MGEs), as abundant resources and high proximity due to high bacterial densities promote horizontal gene transfer of mobile ARGs.”

R1.11: L114: the authors above suggest atmospheric deposition, is this to mean no point and nonpoint source of solid or liquid waste discharges?

Response: Yes. This has now been added for clarification (L.119).

Methods

R1.12: L426-7/442-43: How could one know a priori that samples would have a range of diversities? Was soil/sediment type (i.e., organic matter, sand-silt-clay, conductivity, etc. controlled for?) These parameters do not appear to have been measured and themselves are major drivers of diversity, no?

Response: As mentioned in the comment above, we did not have a priori knowledge due to the exploratory nature of the original sampling campaign. To ensure that this is not wrongfully conveyed we have slightly adapted the wording in this section of the manuscript.

R1.13: L442/443: “not exposed to WWTP” -> “no upstream wastewater treatment plant discharges”. Confirm no upstream agricultural activity, septic systems. Also note the proximity to recreational areas with latrines or human recreational use areas

Response: We have adapted the wording according to the reviewers suggestion and further checked the additional suggested parameters with regards to our sample origins (L. 515-518):

“The locations were selected to obtain samples across a gradient of high and low microbial diversity that are of relatively low anthropogenic impact (e.g., no upstream wastewater treatment plant discharges; no known upstream discharge through agricultural activities or septic systems; no discharge through human recreational areas in the immediate proximity).”

R1.14: Table S2: Provide estimates of stream flow volume and/or widths/depths be provided

Response: As mentioned above, metadata was only sparsely collected, hence it was not possible to retrospectively add this information, as stream flow and depth are changing with times and seasons.

R1.15: Li445: O2 was measured? Or based on other field observations of mixing/lack of methane odors?

Response: Sampling locations were chosen based on field observations, which has now been added to this section of the manuscript (L. 520)

R1.16: L449: shadowed = shaded?

Response: This has been corrected (L.523)

R1.17: Please describe field blanks and frequency

Response: At least one field blank for biofilm collection consisting of sterile water mixed with one of the sterile toothbrushes as well as one extraction blank for each matrix was used in every country. This is now mentioned in the manuscript. For soil and river sediment no field blanks were used as no additional chemicals or solutions were part of the sampling procedure, while extraction blanks were similarly used (L. 508-510 & 533-536).

R1.18: L487: specify quantification limit, describe calibration curve (efficiency, R2). presumably all samples run in triplicate (technical replicate)?

Response: The quantification limit (25 gene copies per reaction) as well as performing three technical replicates are now reported in the methods section. However, as the chip-based HT-qPCR analysis works based on the delta Ct method (Ct of the target gene relative to the Ct of the 16S rRNA gene) rather than a standard curve, no calibration curve is performed and can thus also not be reported (L. 562-566):

“A cycle threshold (CT) of 31 was selected as the detection limit^{39,71}. The quantification limit was calculated as 25 gene copies per reaction accounting for 12.5 gene copies per ng of DNA template. Amplicons with non-specific melting curves or multiple peaks were excluded. The relative abundances of the detected gene to 16S rRNA gene were estimated using the Δ CT method based on mean CTs of three technical replicates⁷².”

R1.19: L145-6: Would this be expected? Geographic borders may not coincide with differences in important environmental parameters such as river flow rate or forest type or soil type or temperature or source water (e.g., snow melt vs groundwater...can no other drivers/factors be explored in these data sets?

Response: We agree with the reviewer that we would not have expected a specific country-based effects and have hence decided to remove the two statements regarding country of origin from this and the previous paragraph. As stated above, due to the original sampling design, other drivers/factors were outside the original sampling scope of this study and could hence not be retrospectively analysed.

Results/Discussion

R1.20: Section beginning L163 – please correct ARG notation – convention not all italics for gene names, only first 3 letters

Response: We have corrected the ARG notation throughout the manuscript and in all figures.

R1.21: Section beginning L195: The significance of these comparisons is not clear to the reviewer.

Response: We have, also in agreement with a comment by reviewer 2, removed this section from the revised manuscript as it did not majorly contribute to the overall story of the manuscript, also for being able to include a more detailed phylogenetic analysis.

R1.22: L228-262: While 27 ARGs is an excellent panel for PCR, there are hundreds detected via metagenomics, so testing correlations of total presence/absence of this subset is not fully justified. The fact that these correlations are generally not maintained when testing an individual ARG also seems to undermine the hypothesis. Were these correlations also tested with the crAssphage marker?

Response: We agree with the reviewer on the limited number of detectable targets obtained through our PCR panel compared to metagenomics. An advantage is the far lower limit of detection through PCR making presence/absence a more reliable measure than when using metagenomics, where sequencing

depth plays a major role in detectability. Sadly, a gold standard method is yet not available for this type of analysis.

However, unlike stated by the reviewer, correlations were generally maintained when testing individual ARGs. In the river dataset where no significant correlation of presence/absence with diversity was detected, equally, no correlations on individual ARG levels were observed. In the soil dataset, where for Shannon diversity and Pielou Evenness negative correlation of presence/absence was observed, equally, for the majority of ARGs on the individual level a negative correlation was observed. Finally, for Chao1 richness, where the correlation was slightly above the significance threshold of negative correlation for presence/absence, at the individual ARG level negative correlations for a number, but not all ARGs were observed. Hence, ARG numbers and individual ARG abundances are indeed displaying identical trends and supporting the hypothesis.

Correlations with crAssphage were also undertaken and a figure added to the SI, even if the number of samples with detectable crAssphage was only 22% in the river dataset. Contrary for the soil dataset this was impossible, as not a single sample displayed any detectable crAssphage. The following statement has now been added to the manuscript (L. 287-291):

“No significant correlation of the observed ARG number or the relative abundance of any ARGs with the relative abundance of crAssphage was obtained for the river dataset (all $p > 0.05$, Spearman, Fig. S3) while crAssphage was entirely absent in all samples of the soil dataset. Thus, it again demonstrates that results are not directly impacted by recent anthropogenic fecal pollution.”

Figure S3. Correlation analysis of ARG abundance with observed crAssphage abundance based on Spearman rank correlation with Bonferroni correction for multiple testing. Correlations from river environmental samples are displayed. For soil samples no correlation was possible as crAssphage was not observed in any of the samples. Filled bars represent significant, while hatched bars represent non-significant correlations.

R1.23: L287: Team demonstrates high diversity has less presence of the 27 ARGs tested, but “resistance to pervasiveness” is not clear and the latter part of this statement is one speculation as to why this was seen

Response: We have removed the statement regarding “resistance to pervasiveness” and clarified that the latter part is indicated rather than demonstrated in the first sentence of the discussion (L. 336-339):
“Here we demonstrate based on analysis of a pan-European sampling campaign that communities of high bacterial diversity display lower diversity and abundance of ARGs, which provides indication that diversity might serve as a barrier to the long-term immigration and establishment of ARGs into environmental endemic microbiomes.”

R1.24: L296: The repetition of the “more dynamic river environment” needs justification – observations here were for single sampling events

Response: We have reformulated the sentence and added a reference for justification (L.343-346):

“While this possible effect of community diversity on long-term invasion and establishment of resistant bacteria was highly visible and frequently statistically significant in the structured soil environment, it was barely observed in the more dynamic river environment characterized by more frequent mixing events and bacterial community succession (Lyautey et al., 2005).”

R1.25: Paragraph L297-399: Quite speculative – no events where invasion could occur were documented...this information can be included in abbreviated form in the discussion, but mostly the study design here isn’t appropriate to answer questions about invasion as no invasion events were documented. This is a missed opportunity to instead a comparison to other pristine environments and ecological drivers

Response: Please refer to our detailed response to comment **R1.3**, which addresses this notion.

Responses to Reviewer #2:

R2.1: Summary and general comments:

The manuscript by Klümper et al. uses a unique suite of soil and river sediment & biofilm samples from across Europe to address their hypothesis that bacterial diversity helps inhibit the ‘invasion’ of antibiotic resistance genes (ARG) into the environment, with implications for the propagation of anthropogenically-sourced ARG inputs via wastewater effluents or application of manure in agricultural systems. The authors utilize the aforementioned set of soil and river samples that span bacterial diversities (via 16S rRNA gene sequencing) to both compare systems that vary in stability and identify intrasystem influences of diversity on ARGs (identified via qPCR). Overall, I believe that the data are solid and that this is an impactful contribution to the research community for assessing future ecosystem risk to anthropogenic ARG inputs – and such a cool story with a neat dataset! However, I think that there are some improvements that should be made to refine the manuscript and further validate the findings & explore the dataset for final publication.

Response: We thank the reviewer for their constructive comments and positive assessment of our study.

R2.2: While the data is comprehensive and is generally overviewed well within the text, I think there is some data and results that are not discussed as thoroughly as needed. I want to note that this dataset is super cool and unique & there is a LOT you could do with it – so I try to only point out things here that I believe would add to the story. One such piece of data was the 16S rRNA gene sequencing data, which was only used to assess diversity. It would be interesting to see whether there are any specific taxa that were any taxa-ARG correlations – i.e., are any bacterial taxa affiliated with the enrichment of a given ARG? Did soil or river samples with a higher abundance of ARGs harbor similar bacterial communities? You use the term ARB – antibiotic resistant bacteria – but then do not discuss any specific taxa that are correlated with ARGs & only briefly mention the phyla-level differences between the soil & rivers, which is already fairly well-characterized. There is also some language in the introduction regarding ‘invaders’ and how, if they are phylogenetically close to the established community, they will have greater success (L82), but again none of the analyses focus on ARBs.

Response: We are now more thoroughly analysing the 16S data in the manuscript, which was indeed a worthwhile endeavour. First, we aimed at identifying potential hosts, to look into if specific ARG hosts are responsible for the observed dynamics in the manuscript, but most ARG abundances were correlated to more than 100 individual ASVs, indicating that it is rather overall diversity than individual hosts that are relevant for the observed effects (L. 209-229):

“Identification of potential ARG hosts

*As a first step of connecting diversity within the samples of each dataset with the ARGs we performed correlation analysis between the obtained ASV and ARG abundances to identify if certain ARGs can clearly be attributed to single or multiple bacterial hosts and would hence potentially be independent of overall community diversity. However, in our datasets spanning geographical distances, no clear host identification was possible: In the soil dataset, each ARGs abundance significantly positively correlated on average with 272 ± 260 ASVs, based on Pearson correlation of abundances with Bonferroni correction for multiple testing (SI Table 1). For 21 of the 25 detected ARGs the number of positively correlated ASVs exceeded 20 and reached up to 984 correlated ASVs for *dfrA*, while for 2 of the ARGs (*aac(3)-VI*, *aph(3’)-Ib*) not a single correlated ASV could be identified. Only for the ARGs *blaCTX-M2* a single ASV (classified as*

Acidobacteria subgroup 2) and for *blaOXA48* two ASVs (classified as *Acidothermus* & *Xanthobacteraceae*) were correlated positively as potential main hosts (SI Table 1). Similarly for the river dataset (SI Table 2), each ARGs abundance significantly positively correlated with 101 ± 88 individual ASVs and exclusively for *vanA* only two potential host ASVs (classified as *Saprospiraceae* and *Sphingobacteriales* AKYH767) were identified through correlation analysis (SI Table 2). However, neither of the potential hosts identified for *blaCTX-M2*, *blaOXA48* in soil and *vanA* in rivers have previously been reported in any of the literature as hosts of these ARGs, in this case correlation is likely not associated with causation. Consequently, ARG abundance in these low-anthropogenic-impact datasets is likely not connected to single hosts, allowing for subsequent analysis if overall community diversity is the hypothesized predictor of ARG abundance.”

Second, we now explore if total ARG abundance in the samples is predictable through phylogenetic similarity of the samples. Here we found that samples with high ARG abundance are indeed phylogenetically more similar, while those that display low abundance are more dissimilar when comparing them to the entire dataset (L.309-334):

“Similarity of communities according to total ARG abundance

Finally, we aimed at establishing if aside from community diversity, the abundance of ARGs in a sample is also predictable through phylogenetic similarity with samples of similar ARG contents. To achieve this, each of our two datasets was divided into three subsets: 1) the full dataset, 2) those 20% of samples with the highest total ARG abundance (Top 20%) and 3) those 20% of samples with the lowest total ARG abundance (Bottom 20%). Across both datasets, the samples of the Top 20% in total ARG abundance subset displayed significantly lower average pairwise Bray Curtis dissimilarity, and hence a higher degree of similarity, with each other compared to all samples in the full dataset (Fig. 7). In the river dataset dissimilarity decreased from 0.916 ± 0.089 ($n = 4186$ pairwise comparisons) in the full dataset to 0.873 ± 0.130 in the Top 20% of samples ($n = 153$) ($p=0.0066$, one-way ANOVA with Tukey HSD). Similar in the soil dataset, dissimilarity decreased from 0.867 ± 0.109 (full dataset, $n=2016$) to 0.821 ± 0.147 (Top 20%, $n=66$) ($p=0.0199$) (Fig. 7). Contrary, those 20% of samples with the lowest total ARG abundance displayed throughout a higher dissimilarity with each other than both the full sample dataset and the Top 20% dataset (all $p < 0.05$; Fig. 7). Consequently, while community diversity is correlated with ARG abundance in at least the structured soil environment, community similarity can only serve as a predictor for high ARG abundances, but is a bad predictor of low ARG abundance as low abundance samples have a high degree of dissimilarity.”

Figure 6. Distribution of pairwise Bray Curtis dissimilarities between samples in the river and the soil dataset. Distribution of pairwise dissimilarities for each dataset is displayed across all samples within the dataset as well as across those subsets of samples with the 20% lowest (Bottom 20%) and the 20% highest (Top 20%) total ARG abundance. Significance testing between pairwise dissimilarity distributions is performed through one-way ANOVA with post-hoc Tukey HSD.

This is now also discussed at the end of the discussion (L465-474):

“Finally, we tested if similarity in ARG abundances is predictable through phylogenetic similarity of the hosting microbiomes. Here, it became apparent that communities with high total ARG abundances are indeed phylogenetically closer to one another than would randomly be expected from the entire dataset. However, communities with low ARG abundances were even more dissimilar than the average samples dissimilarity in the entire dataset. This indicates that the ability to host ARGs at high abundances is rather a specialist trait and hence manifests in the phylogenetic composition of the community 65,66. Contrary, low ARG abundances are not based on the phylogenetic composition of the community, but rather on the above demonstrated diversity-based barrier effects and can hence be characterized as a generalist community trait.”

R2.3: Another piece of data that I feel wasn’t discussed enough was the abundances of blaTEM and how different its profile was compared to other surveyed genes (e.g., consistently high abundance in Ireland sites and it being the only gene with a significant positive correlation to diversity in both systems. This clear distinction between it and the other ARGs kept popping up to me as a reader and it needs to be discussed more than just pointing it out – is there any reason why this ARG would act so differently to the others? It especially needs to be discussed because it is consistently so different. Further, I think there could be more discussion on specific ARGs, like why is there not a significant correlation with some of them (e.g., dfrA1, aac(6′)_ib, etc.)?

Response: We agree that the case of blaTEM is a curious one and deserves further attention. In accordance with other comments by the reviewers we have decided to focus our discussion rather on the reasons why

this gene is acting so different than on a particular discussion regarding geography based on Irish samples. Based on previous reports (cited in the manuscript now, see below), bla_{TEM} is a particularly bad indicator of any short- or long-term anthropogenic impact and is often reported at higher abundance and an integral part of the natural resistome in control soils or aquatic samples compared to those impacted through anthropogenic activity. So unlike for the other ARGs its origin is likely not through the suggested enrichment in the anthroposphere and subsequent invasion into the environment, explaining its contrary behaviour. This is now discussed in the manuscript (L.427-434):

“An interesting observation in this context was that the beta-lactam ARG bla_{TEM} displays an opposing behaviour to all other ARGs as its abundance was rather positively correlated with diversity in both datasets. In previous studies, bla_{TEM} was regularly found to not correlate with other ARGs, crAssphage as an anthropogenic pollution indicator, or general introduction of solid or liquid human associated waste in both, soil^{12,51} and aquatic^{10,52} environments. Rather it was found at higher abundance in non-impacted environments as an integral part of the natural resistome^{12,51,52}. Consequently, unlike for the majority of the other gene its main origin is likely not from anthropogenic enrichment and subsequent invasion over long time periods.”

We have however decided against a more specific discussion of other individual ARGs, as we feel that such a discussion would be highly speculative.

R2.4: Something else that was briefly mentioned but not discussed as thoroughly as it should have been was the mobile genetic element (MGE) data. You first mention them in L275, then briefly discuss them in the discussion (~L390). I also see them in Table S3, but do not see any of the actual abundance data from them in Table S1 or S2. If this data is going to be included in the manuscript, the data needs to be included in Table S1 and S2 and you need to include a SOM figure showing the lack of correlation between MGEs and diversity. This is another cool piece of data that adds to the story but I don't feel is utilized enough.

Response: We particularly thank the reviewer for bringing this up as we have found a mistake in our original code that was used to compute the MGE correlations when creating the SOM figure for this. We have now been able to correct this and the new corrected analysis demonstrates that MGEs do indeed follow a similar trend of negative correlation with diversity in the soil dataset. Data is now integrated into Table S3 and S3, a figure and corrected results have been integrated into the results section, and the discussion has been adjusted to mirror the corrected results (L. 281-291 & L.444-464):

“Similar to ARGs, four of the five indicator genes for MGEs quantified in parallel through high-throughput qPCR (the class1 integron integrase gene intI1, the IncP plasmid oriT, the IncW plasmid trwAB gene, the orf37 of IS26) displayed negative correlation with all three diversity indices in the soil dataset (all p>0.05), while no effect for the Tn5 transposase gene was observed (Fig. 5 D-F). Again, in the river dataset, no or slightly positive correlations of MGE abundance with the diversity indices were observed, mirroring the effects on ARG abundance (Fig. 5 A-C).”

Figure 5. Correlation analysis of relative MGE abundance with observed diversity metrics based on Spearman rank correlation with Bonferroni correction for multiple testing. Correlations from river environmental samples with Pielou Evenness (A), Shannon Diversity (B) and Chao1 Richness (C). Correlations from soil environmental samples with Pielou Evenness (D), Shannon Diversity (E) and Chao1 Richness (F). Filled bars represent significant, while hatched bars represent non-significant correlations.

“However, effects of community diversity on the efficiency of horizontal gene transfer and the maintenance of plasmids in the community consist of a complex interplay of different mechanisms and remain difficult to predict. On the one hand, at higher diversity an increased number of potential plasmid hosts and conjugation partners are available that can lead to increased plasmid maintenance and transferability in the community^{58,59} increasing the chance of transfer to a highly competitive host. On the other hand, in more diverse communities it can be harder to encounter a permissive conjugation partner, which reduces transferability due to this dilution effect⁶⁰. Further, competition with other community members might increase the costs of resistance⁶¹ and could ultimately drive the loss of ARG hosting plasmids from the community⁶². This loss process would be expected to be elevated in more diverse communities with better competitors. Our dataset provides a good indication, that it is rather the latter processes that are dominant in structured environmental communities as in our soil dataset, similar to ARGs a clear negative correlation for four of the five MGEs with diversity could be established. Hence, community diversity might also limit the horizontal acquisition of mobile ARGs from invading bacteria ultimately resulting in lower numbers and abundances of detected ARGs in the soil dataset. This is according to ecological theory, where species diversity is not always immediately implying a higher degree of genetic diversity^{63,64}. Still, assuming that, in the long-term, invaders harboring the tested ARGs reach each of the tested communities it becomes apparent that an increasing number of ARGs are not successfully retained in those communities of higher diversity. If this is due to a shorter residence time of the invader, the above discussed increased competition, decreased horizontal gene transfer potential or dilution effects needs future research.”

R2.5: Along with these pieces of data that I feel could be more thoroughly discussed or intertwined throughout the manuscript, I did not feel like that ‘Higher degree of correlation between relative ARG abundances in soils compared to rivers’, along with Figure 3, added much to the story. I just didn’t fully follow how the correlation between specific ARG abundances fit or informed the hypothesis. I may be completely missing something (& please push back if so!) & this language needs to be tightened and made clearer, or this section could be shortened to make room for some of the other analyses I suggested above.

Response: We agree with this and reviewer 1’s comment regarding this paragraph and figure not being necessary to the story and have hence decided to remove it to make the room for the suggested phylogenetic analyses.

R2.6: Lastly, is there any chemistry metadata on any of these samples, specifically the soil samples? It seems odd to have all of these samples and only the microbial data, when we know that in environmental systems the chemistry and microbiology have complex feedbacks, which likely also influence ARGs. This might be out of scope of this study & might be data that’s being used somewhere else but, as a soil microbiologist, we rarely publish soil microbiome data without including chemistry metadata.

Response: This limitation of the study has also been pointed out by reviewer 1. Both reviewers are right and we are aware of this. The mentioned limitation is based on the origin story of the dataset: The sampling campaign was originally carried out to screen a large number of soils and river biofilm/sediment samples in order to identify suitable soils/rivers of differing diversities for short-term laboratory invasion experiments with resistant bacteria (e.g., <https://doi.org/10.1016/j.scitotenv.2023.166661>). Hence, only a limited amount of metadata was collected as sample collection was carried out in exploratory manner in hopes of gaining a high variety of diversity levels rather than actively pursuing it through a priori metadata evaluation. A main reason for this was the very differently equipped project partners in the 7 different countries with expertise and specialization on different matrices.

However, once we realized that the captured diversity in the dataset was indeed highly variable, we were wondering if diversity would not only affect the short-term invasion of resistant bacteria (as mentioned above), but also affect long-term levels of resistance in the community, which we address through the analysis presented in this manuscript with high effect sizes for the soil samples.

To further address this comment, we have now collected as much metadata as was available for the samples based on previous or retrospectively possible analysis of samples available through the different partners and added it to the SI tables. Still, the available metadata remains largely heterogenous across sample type and countries. While we now tested if any of the soil or water properties significantly affected ARG levels, no significant patterns emerged. This could be either due to them not existing or due to the lack of data and hence statistical power for certain parameters for a large proportion of samples. As this additional analysis was not very informative due to the restrictions mentioned above, we have hence decided to not include it in the revised manuscript, but rather to only provide the available metadata in the SI tables.

R2.7: L92 – ‘as this usually coincides with a lower rate of niche occupation’, this needs a reference. I know you cite references that would work here above, but something needs to be here. “Usually coincides..” is too arm-wavy of a statement to not be followed by a citation that supports

Response: We have added the relevant citation (L.94).

R2.8: L99 – L108 – Just want to say that I love this paragraph! It frames the study nicely.

Response: Thank you!

R2.9: L112 – The use of ‘long-term’ should be removed here, it suggests a time element to this study.

Response: long-term has been removed accordingly.

R2.10: L122 & L125 – Examples of inconsistent tense used here (e.g., ‘while the other half was obtained’ should be ‘while the other half were obtained’). Keep tense consistent throughout text.

Response: This has been corrected and we went carefully through the entire manuscript to correct any additional inconsistencies.

R2.11: L124 – Introduce the term resistome here (e.g., ‘microbial diversity was assessed through 16S rRNA gene-based amplicon sequence, while the resistome was analyzed via the abundance of 27...’). This term is just dropped in the results section but never explained and some readers might not be familiar.

Response: We have now introduced the term resistome here and also untangled the very long sentence, while also already introducing the quantification of MGEs and crAssphage in this section, as now according to the remaining comments, all correlations for these markers are also part of the manuscript (L. 129-135):

“The resistome, defined as the collection of all ARGs in a microbiome, was analyzed via abundance of 27 clinically-relevant ARGs determined through high-throughput chip-based qPCR. Simultaneously, the abundance of mobile genetic elements (MGEs) in samples was assessed through 5 marker genes regularly associated with AMR and the anthropogenic fecal pollution indicator crAssphage^{32,33} was quantified.”

R2.12: L124 – ‘bacterial 16S rRNA gene-based amplicon sequencing’ should be changed to just ‘16S rRNA gene sequencing’. The fact that its 16S implies that its bacterial, and ‘microbial diversity’ should be changed to ‘bacterial diversity’ because microbial could include fungi & viruses along with bacteria.

Response: We have corrected this at this instance, and went carefully through the entire manuscript to ensure that at those instances where only bacteria were analysed the term bacterial diversity is used.

R2.13: L127 – I recommend moving the hypothesis statement to the end of this paragraph to leave the reader clearly knowing before going into results.

Response: While we prefer to keep introducing the general hypothesis at the beginning of the paragraph based on the theory presented before, we agree that a clearer statement at the end of the paragraph was needed. We hence now refer back to the testing of the hypothesis in the final sentence of the introduction (L-134-137):

“This allowed to ultimately test the previously presented hypothesis that AMR in low-anthropogenic-impacted environmental microbiomes is inversely correlated to the diversity of the communities in question as microbiome diversity can serve as a barrier against the spread of ARGs.”

R2.14: L131 – Because the methods are at the end in this journal format, I’d make sure to nod towards the methods throughout the results section. E.g., here it is unclear what the ‘river’ samples are. Be clear that these are not surface water samples, but are river sediment & biofilm samples. The use of the term river is very vague.

Response: We have now clarified from the very beginning, that the river dataset consists of river sediment and river biofilm samples and given the corresponding numbers of samples for each sample type immediately (see also the response to the comment below referring to the n values) (L.140-146):

“Two complementary sets of samples of low anthropogenic impact were obtained from a total of 94 riverbed (61 river epilithic biofilm and 33 river sediment samples) as well as 73 soil samples. When assessing the beta diversity of the bacterial communities, no clear distinction was observed comparing sediments and epilithic biofilms (PERMANOVA, pseudo-F = 2.77, $p > 0.05$). Consequently, these samples were subsequently grouped to create the combined river dataset. Soil samples differed significantly and with a large effect size from those obtained from river samples (PERMANOVA, pseudo-F = 14.73, $p < 0.001$) (Fig. 1A).”

R2.15: L132 – Again, instead of ‘microbial communities’ I’d use ‘bacterial communities’ so as not to imply the inclusion of fungi or viruses. Folks especially think about fungi in soils and could misinterpret the term as inclusive of fungi.

Response: As stated above, we have carefully vetted the manuscript to clearly state that all presented results are based on bacterial rather than microbial diversity. We are actually considering including fungal diversity in a follow up study, as it is of particular importance in at least the soil environment.

R2.16: L134 – L135 – Why are these n values so much larger than the total number of samples from both systems that are mentioned in the first sentence of this paragraph? I don’t understand how you have 148 river biofilm samples when you mention 94 in L131. Did this mean 94 rivers or samples? Be very clear here, as the methods aren’t until the end of the manuscript and you don’t want readers to have to search.

Response: The n values originally referred to the combined number of samples in each comparison (e.g. soils + river biofilms or river biofilms + river sediments). However, as this was indeed confusing, we have rephrased this paragraph to first introduce the number of samples for each type and thereafter present the statistical comparisons without the need to restate the n values for each comparison (L.140-146):

“Two complementary sets of samples were obtained from a total of 94 river (61 river epilithic biofilm and 23 river sediment samples) as well as 73 soil samples. When assessing the beta diversity of the bacterial communities, no clear distinction was observed comparing sediments and epilithic biofilms (PERMANOVA, pseudo-F = 2.77, $p > 0.05$). Consequently, these samples were subsequently grouped to create the combined river dataset. Soil samples differed significantly and with a large effect size from those obtained from river samples (PERMANOVA, pseudo-F = 14.73, $p < 0.001$) (Fig. 1A).”

R2.17: Fig 1A – There’s a lot going on here, I’d add some ellipses around the samples representing different substrates to help the reader see the differences between river & soil samples that you mention in the text.

Response: We have added the suggested ellipses to the figure and explained how they were created in the figure legend:

Figure 1. Diversity of the river and soil datasets. Symbols depict sample type, colors code for the country of origin. **A)** PCoA of the beta diversity based on Bray-Curtis distance of ASV relative abundance data from riverbed materials (sediments and biofilms) and soil. Ellipses were drawn based on a 95% confidence interval to represent samples from each of the substrates. **B)** Alpha-diversity indices (Chao1 richness, Shannon diversity and Pielou evenness) from riverbed materials (top) and soil (bottom) collected from the seven countries.

R2.18: L137 – The explanation that some river communities differed between country seems misplaced here and does not add to the story. I recommend removing this note.

Response: We have removed the two mentions of no differences across countries from this section of the manuscript as we agree that it does not add to the story and bacteria do not really care about country borders.

R2.19: L143 & L144 – Change the values to percents (I think they are currently relative abundance out of 1?). You mention relative abundance previously (L141) so should keep it consistent.

Response: Values have been changed to percent for consistency.

R2.20: L143 – Minor but Figure S2 is mentioned before Figure S1, should renumber so that they are in order.

Response: The numbering of Figures and SI Figures has been revised throughout, as new figures have been added.

R2.21: L145 – Whenever you use term ‘significant’ should be along with some sort of statistical test. Did you do a stat test here? If not, remove.

Response: This sentence has been removed in accordance to the previous comment regarding country effects, hence the problem is resolved. We also went through the entire manuscript to ensure that significance is throughout statistically confirmed.

R2.22: L146 – Need a reference with the introduction of CrAssphage.

Response: References have been added here as well as in the introduction, where crAssphage is now mentioned first.

R2.23: Fig 1B – Would be helpful if the river and soil values were plotted next to one another instead of above & below, especially because you directly mention the difference between them.

Response: We now plot the river and soil values next to each other to allow for a better visual comparability. See figure in response to comment R2.17.

R2.24: L163 – In this title, you mention diversity but don’t really discuss diversity of the ARGs. You do a bit later with noting the # of ARGs in different samples, but I think it would be helpful here to include a SOM fig that shows diversity of ARGs between samples.

Response: We have now moved the number of ARGs in different samples to the beginning of this paragraph as it seems better suited here. Furthermore, diversity of ARGs between samples is already displayed in the similarity tree based on Euclidian distance in Figure 2, while within samples diversity is displayed in the heatmap.

R2.25: L164 – Briefly state the methods here before jumping into the data. E.g., ‘We analyzed the resistome of our samples using qPCR of X different genes...’.

Response: Thank you for this suggestion. We have now added the methods statement at the beginning of the paragraph (L. 172-174):

“To analyze the resistome of the soil as well as the river samples we performed high-throughput qPCR of 27 ARGs as well as the 16S rRNA gene to obtain relative abundances of these ARGs for each sample.”

R2.26: L176 – Minor but I’d order these as the figure is ordered. Also, acc(3)-VI needs to be aac(3)-VI.

Response: We have reordered and corrected

R2.27: Figure 2 – Edit the y axes so that they are the polished gene names (like what you have in Fig 5) and increase text size of legend.

Response: The figure has now been revised accordingly:

R2.28: Figure 3 – Again, gene names should match how they are written in Fig 5. Also difficult to read names here. Need to add label legend.

Response: The figure together with the paragraph have been removed from the manuscript.

R2.29: L208 - 211 – This sentence reads confusing to me and is maybe where I got mixed up on the significance of this section and these findings. Please tighten language and relevance of the findings to the hypothesis. Further, state here what kind of network analysis was used (WGCNA?). The reader should have to go to the methods for specific details but not just the type of analysis!

Response: We have, as stated above, entirely removed this part of the manuscript according to the reviewer's suggestion.

R2.30: L212 – Could you show this in a SOM figure?

Response: CrAssphage correlations for rivers are now displayed in an additional figure in the SI, while for soil no correlation figure was possible, as crAssphage was not detected in any of the samples:

Figure S3. Correlation analysis of ARG abundance with observed crAssphage abundance based on Spearman rank correlation with Bonferroni correction for multiple testing. Correlations from river environmental samples are displayed. For soil samples no correlation was possible as crAssphage was not observed in any of the samples. Filled bars represent significant, while hatched bars represent non-significant correlations.

R2.31: L227 - 228 – The statement about the number of ARGs detected in samples in the river (these lines) and soil (L241 – L242) seems out of place here and should be at the beginning of the resistome diversity and abundance section.

Response: We have moved the statement accordingly.

R2.32: Figure 4 – This figure should be plotted the opposite way – the diversity value should be on the x

axis and number of ARGs on the y axis because you're implying that the number of ARGs is the dependent variable here.

Response: The x and y axis have now been swapped:

Figure 3. Correlation analysis of the number of ARGs detected per sample with diversity metrics based on Pearson correlation with Bonferroni correction for multiple testing. Linear correlations from river environmental samples with Pielou Evenness (A), Shannon Diversity (B) and Chao1 Richness (C). Linear correlations from soil environmental samples with Pielou Evenness (D), Shannon Diversity (E) and Chao1 Richness (F). Colors depict the country of sample origin and the symbols depict the sample type.

R2.33: Figure 5 – I really, really love this figure! Elegant way to show a lot of data & tests at once. Make the legend and axes text a bit bigger.

Response: Thank you! Legend and Axes text have been made bigger:

Figure 4. Correlation analysis of relative ARG abundance with observed diversity metrics based on Spearman rank correlation with Bonferroni correction for multiple testing. Correlations from river environmental samples with Pielou Evenness (A), Shannon Diversity (B) and Chao1 Richness (C). Correlations from soil environmental samples with Pielou Evenness (D), Shannon Diversity (E) and Chao1 Richness (F). Filled bars represent significant, while hatched bars represent non-significant correlations. Colors depict the class of antibiotic the ARG confers resistance to. Only ARGs that were detected in at least 25% of samples of a dataset were tested.

R2.34: L275 – Vague here what ‘tested’ means – do you mean targeted via qPCR? Further, this is already mentioned in the major comments section, but the MGE data should be elaborated on! Plus, lack of correlation would be nice to see in SOM fig.

Response: “tested” has been replaced by “quantified in parallel through high-throughput qPCR”. A figure has been added as stated in the response to the comment above **R2.4**.

R2.35: L407 – The use of ‘texture’ here is odd and only applies to soil & river sediment, not so much the river biofilms. Plus, in the discussion you explain some of the dynamic nature of rivers (L361-363) but don’t talk about soils much. While soils can be thought of as more stable in time than river environments, I think it would help to give a nod to the fact that soils are very heterogenous in space as

compared to time and how this may influence your findings or the implications of this on the proliferation of ARGs in soils.

Response: We have removed the term texture here, as it did not particularly contribute to the message. We have furthermore added the nod to spatial heterogeneity in the discussion section (L. 423-426):

“However, while soils can be thought of as more stable in time than river environments, soils are highly heterogenous in space (Baveye and Laba, 2014), meaning that the strength of the observed barrier effect could also be highly variable on the spatial scale if niche occupation varies.”

R2.36: L439/sampling details – You should be very clear throughout the text that the river samples are sediment and biofilms, not surface water. As an environmental micro person, I was a bit confused at the beginning of the manuscript what ‘river samples’ entailed. Make sure you’re very specific throughout the entirety of the text so the reader knows what environment you’re discussing.

Response: This has now been clearly pointed out at the beginning of the results section to avoid any confusion to the reader.

REVIEWERS' COMMENTS:

Reviewer #1 (Remarks to the Author):

L271-5: correlation also may not be the correct way to do this- depending on how annotations were done (eg, 16S which we know has varying numbers of copies by organism and life phase). This is why you see folks comparing this network type analysis with assembly and other methods that would deserve mention

L278: again, since no inputs were measured here the study design isn't appropriate for this type of statement regarding "invasion". If you don't know the inputs how can you assess their persistence? L341-5 also would seem to indicate there weren't locations with recent invasion opportunities and therefore may not be good for measuring this invasion hypothesis. The reviewer takes no issue with this comparative study beyond the authors use of it as an opportunity to test an invasion hypothesis without supporting data demonstrating inputs

The relationship / lack thereof of for ARG in diverse communities is finely substantiated

L405-7: wouldn't antibiotics also impact the diversity? It would seem a bit more nuanced an effect rather than an either/or

L475-86. But what of the fact that several antibiotics were isolated from soil microbes? These chemicals play a role in signaling and/or competition among soil microbes that was then taken into clinical settings...potentially different from aquatic microbes

Reviewer #2 (Remarks to the Author):

The authors have fully addressed my comments from the first round of reviews and I commend them on their thoughtful & thorough responses to previous comments and subsequent alterations to the manuscript. I also very much understand (& lament with!) the lack of corresponding consistent sample metadata due to the nature of the sampling campaign and varying project partners ability and think that, for the purpose & hypothesis of this manuscript, this is fine and does not present a major issue. I would add in one sentence to the methods (maybe the "Soil sampling and processing" section) that mentions that there is some corresponding metadata included in Table SX but that there were limitations in further analyses with these due to the nature of the sampling campaign. This might be someplace, but I can't find it. Overall, I believe that this manuscript is greatly improved and have no further comments or suggestions.

We thank the editor and the two reviewers for their time, dedication and helpful assessment of our manuscript. We have now implemented the final requests by the reviewers. Please find below a detailed response to each individual reviewer comment (numbered to allow cross-referencing). Responses are throughout given in blue, with relevant changes in the revised manuscript cited in *blue italics*. Additionally line numbers referring to the new clean version of the manuscript are referenced for easy checking of any edits, additions, etc..

Responses to Reviewer #1:

1. L271-5: correlation also may not be the correct way to do this- depending on how annotations were done (eg, 16S which we know has varying numbers of copies by organism and life phase). This is why you see folks comparing this network type analysis with assembly and other methods that would deserve mention

Response: We in general agree with the reviewer that metagenomic analysis with assembly and network analysis could resolve this issue to a higher degree. This is regularly carried out for anthropogenically impacted environments, and could provide a clearer picture of exact hosts. However, taking into account the low detection limit of metagenomics, the high diversity of potential host bacteria and low relative abundance of ARGs in the samples, the necessary sequencing depth to capture individual ARG-host relationships was considered disproportionate for this study, where ARG-host relationships play only a minor role to the overall story. Still we now qualify the choice of method in the respective paragraph:

L. 209-219:

“Metagenomic analysis with contig assembly and network analysis, as regularly carried out for anthropogenically impacted environments with high ARG abundance and lower bacterial diversity^{31,34}, might be able to provide a higher resolution of exact hosts. However, taking into account the low detection limit of metagenomics, the high diversity of potential host bacteria and generally low relative abundance of individual ARGs in these low impacted environmental samples, the necessary sequencing depth and coverage to conclusively capture the potential ARG-host relationships was considered disproportionate. Consequently, based on the applied correlation analysis we conclude that ARG abundance in these low-anthropogenic-impact datasets is likely not connected to individual but rather multiple hosts, allowing for subsequent analysis if overall community diversity is the hypothesized predictor of ARG abundance.”

2. L278: again, since no inputs were measured here the study design isn't appropriate for this type of statement regarding “invasion”. If you don't know the inputs how can you assess their persistence?

Response: We have here, and throughout the manuscript toned down the statements that we directly test for invasion success especially (as pointed out by the reviewer) in the results section where these statements were particularly too strong. We still believe that discussing our results in the framework of immigration/invasion theory remains necessary though as it is one of the most important mechanisms underlying the spread of AMR in environmental communities. Still, also in the introduction and discussion we now avoid any statements that (long-term) invasion is the exclusive mechanism that can explain the observed dynamics.

3. L341-5 also would seem to indicate there weren't locations with recent invasion opportunities and therefore may not be good for measuring this invasion hypothesis. The reviewer takes no issue with this comparative study beyond the authors use of it as an opportunity to test an invasion hypothesis without supporting data demonstrating inputs. The relationship / lack thereof of for ARG in diverse communities is finely substantiated

Response: The reviewer is right, that no recent invasion opportunities materialized. As mentioned in the comment above (2.) we have hence toned down the claim that we are exclusively testing for the invasion hypothesis.

4. L405-7: wouldn't antibiotics also impact the diversity? It would seem a bit more nuanced an effect rather than an either/or

Response: The reviewer is right, and we have now implemented this more nuanced effect in L. 307-312:

"If, rather than a general diversity effect on ARG spread, other ecological drivers (e.g., the presence of chemical stressors) would be responsible, a more diverse effect would have been expected as the presence of such stressors would, while reducing overall diversity through inhibiting certain community members, particularly favor selection or co-selection of specific, individual ARGs."

5. L475-86. But what of the fact that several antibiotics were isolated from soil microbes? These chemicals play a role in signaling and/or competition among soil microbes that was then taken into clinical settings...potentially different from aquatic microbes

Response: We have included this important notion into the revised version of the manuscript L. 381-390:

"Moreover, the competitive ability of soil microbes against foreign bacteria might be elevated compared to those occupying riverbeds as several indigenous soil bacteria are known producers of antimicrobial compounds involved in competition and cell-to-cell signaling^{50,51}. Less is known if such antimicrobial producers also occur in river biofilms at equal abundances, still any compounds produced would only transiently provide an advantage before being washed away due to the rivers' aquatic nature. Hence, soil microbiomes can be thought of as more stable and more competitive in time than river environments. Still, soils are highly heterogenous in space⁵², meaning that the strength of the here proposed barrier effect could also be highly variable on the spatial scale if niche occupation varies."

Responses to Reviewer #2:

The authors have fully addressed my comments from the first round of reviews and I commend them on their thoughtful & thorough responses to previous comments and subsequent alterations to the manuscript. I also very much understand (& lament with!) the lack of corresponding consistent sample metadata due to the nature of the sampling campaign and varying project partners ability and think that, for the purpose & hypothesis of this manuscript, this is fine and does not present a major issue. I would add in one sentence to the methods (maybe the "Soil sampling and processing" section) that mentions that there is some corresponding metadata included in Table SX but that there were limitations in further

analyses with these due to the nature of the sampling campaign. This might be someplace, but I can't find it. Overall, I believe that this manuscript is greatly improved and have no further comments or suggestions.

Response: We thank the reviewer again for their second assessment of our manuscript. We have according to their comment added a sentence to the Materials and Methods section regarding metadata for the soil and water sampling processing sections.